# Overview of the Development, Impacts, and Challenges of Live-Attenuated Oral Rotavirus Vaccines

**DOI:** 10.3390/vaccines8030341

**Published:** 2020-06-27

**Authors:** Olufemi Samuel Folorunso, Olihile M. Sebolai

**Affiliations:** Department of Microbial, Biochemical and Food Biotechnology, University of the Free State, PO Box 339, Bloemfontein 9300, South Africa; SebolaiOM@ufs.ac.za

**Keywords:** rotavirus, rotavirus vaccines, vaccine effectiveness, vaccine efficacy, gastroenteritis, diarrhoea, intussusception, hospitalisation, immune-response, seroconversion

## Abstract

Safety, efficacy, and cost-effectiveness are paramount to vaccine development. Following the isolation of rotavirus particles in 1969 and its evidence as an aetiology of severe dehydrating diarrhoea in infants and young children worldwide, the quest to find not only an acceptable and reliable but cost-effective vaccine has continued until now. Four live-attenuated oral rotavirus vaccines (LAORoVs) (Rotarix^®^, RotaTeq^®^, Rotavac^®^, and RotaSIIL^®^) have been developed and licensed to be used against all forms of rotavirus-associated infection. The efficacy of these vaccines is more obvious in the high-income countries (HIC) compared with the low- to middle-income countries (LMICs); however, the impact is far exceeding in the low-income countries (LICs). Despite the rotavirus vaccine efficacy and effectiveness, more than 90 countries (mostly Asia, America, and Europe) are yet to implement any of these vaccines. Implementation of these vaccines has continued to suffer a setback in these countries due to the vaccine cost, policy, discharging of strategic preventive measures, and infrastructures. This review reappraises the impacts and effectiveness of the current live-attenuated oral rotavirus vaccines from many representative countries of the globe. It examines the problems associated with the low efficacy of these vaccines and the way forward. Lastly, forefront efforts put forward to develop initial procedures for oral rotavirus vaccines were examined and re-connected to today vaccines.

## 1. Introduction—Overview of Rotavirus Particles

Rotavirus (RV) is an icosahedral non-enveloped Reoviridae family member with six structural proteins called viral proteins (VP) and six non-structural proteins (NSP). The 18.5 kb genome is made up of 11 segmented dsRNA [1,2]. Each of these genes possesses at least one long open reading frame (ORF) after a strong starting codon (based on Kozak’s rule) [3]. While segment 9 (and sometimes 7 and 10) has an additional in-phase ORF, segment 11 has an out-of-phase ORF. All the gene segments are monocistronic [4,5] except segment 11, which is currently suspected to be tricistronic, though only two polypeptides are known for this segment [6,7].

Structurally, rotavirus can exist in three different forms; the single-layered particle (SLP), which consists of the VP2 enclosing the genomic materials and scaffolding the transcriptional enzymes, the double-layered particle (DLP), which consists of the VP6 enclosing the VP2, and the triple-layered particle (TLP), which consists of the VP7 that encloses the VP6 and VP2 [8] (Figure 1). The VP4 are protease-sensitive spike-like proteins that intercalate the VP7 and form the bases for the P-type classification, while the glycosylated VP7 forms the bases for the G-type classification [9]. Among the nine antigenic groups identified (RVA–RVI) based on the antibody cross-reactivity with the conserved region of antigenic capsid protein, VP6 [10,11], rotavirus group A (RVA) remains the most important group responsible for more than 90% of the human gastrointestinal infections caused by rotavirus. It is also the most widely studied rotavirus group, which currently has 36 G-types and 51 P-types (https://rega.kuleuven.be/cev/viralmetagenomics/virus-classification).

## 2. Rotavirus Vaccines—Implementation, Efficiency, Cost, and Challenge

The efficiency of live-attenuated oral rotavirus vaccine (LAORoV), as measured by its magnitude of efficacy or effectiveness, is quite remarkable against dehydrating diarrhoea caused by rotavirus infections (RVIs) [14]. The report has shown that diarrhoea is currently the fifth cause of death globally [15]. With the introduction of rotavirus vaccines in 2006, the total number of deaths from diarrhoea decreased by 16.6% between 2007 and 2017, with 1.88 million to 1.57 million deaths. Consequently, the death rate as a result of diarrhoea decreases from 31/100,000 in 2007 to 21.6/100,000 in 2017. Specifically, within this range of years, the mortality rate from diarrhoea decreased by 40.6% in children <5 years [15]. Regarding RVI, the mortality rate in children <5 years declined sequentially from 453,000 in 2008 [16] to a putative 128,500 in 2016 [17]. This is possible because of the concurrent efforts of the two globally licensed live-attenuated oral vaccines, Rotarix^®^ (GlaxoSmithKline Biologicals, GSK, Rixensart, Belgium) and RotaTeq^®^ (Merck & Co. Inc., Whitehouse Station, NJ, USA), first introduced in the USA by 2006. Recently, two Indian-produced live-attenuated oral vaccines named Rotavac^®^ (Bharat Biotech, Hyderabad, India) and RotaSIIL^®^ (Serum Institute of India Pvt. Ltd., Pune, India) have been pre-licensed in January 2018 and September 2018, respectively, to be used globally against RVIs [14]. However, the use of these recently licensed rotavirus vaccines is still largely restricted to India and partly Africa; perhaps the focus is on the LMICs. Apart from these, there are other regionally licensed, LAORoVs such as LLR-37 from Lanzhou Biologicals (Chengguan District, Lanzhou, Gansu, China) or Xinkexian Biological Technology (Fuyang, Anhui, China) [18] and Rotavin-M1^®^ from Polyvac (Center for Research and Production of Vaccines and Biologicals), Hanoi, Vietnam [19].

Higher rotavirus vaccine (RoV) efficacies have been reported in high- and middle-income countries (HMICs) than in LICs [20,21]. Nevertheless, the impact of these vaccines, as recorded in Africa and Asia, is associated with a vast reduction in the mortality rate of severe gastroenteritis (SGE) and all forms of hospitalised-diarrhoea in children and infants [22]. Reports from RoV surveillance studies attributed the lower efficacy to several factors. These factors include the nature of the circulating rotavirus strains [23], co-administration of the vaccine with oral polio vaccine (OPV) [24], transplacentally acquired maternal rotavirus-specific antibodies in infant [25,26], co-infections (such as norovirus and enteric bacterial infections) [27], microbiota [28], body immune-status, and general nutritional status [29,30]. RVI is contagious through the faecal-oral-route only [31,32]. WHO has continued to reiterate the implementation of rotavirus vaccines (RoVs) as part of the comprehensive strategy to control diarrhoea and also promote strategic prevention (including early and exclusive breastfeeding, handwashing with soap, improved water, and sanitation) and treatment packages (including a low-osmolarity oral rehydrating solution (ORS) and zinc supplementation). This implementation will promote disease-free conditions and improve general wellbeing, and this will indirectly buttress the RoV efficiency because as at the end of 2017, the global RoV coverage was still low, down to about 28% [33].

Despite the outstanding reports of these vaccines from the clinical trials and surveillance studies, it is important to re-emphasise that RVI is still the leading cause of death related to gastroenteritis (GE) in children <5 years worldwide [34]. Currently, there are 107 countries (54.9%) that have introduced the use of RoVs with 103 national and 4 sub-national implementations [35]. With the Global Alliance for Vaccines and Immunisations (GAVI) co-financial support, more than 10 high-burden countries are already planning to introduce RoVs their Expanded Programme on Immunisation (EPI) [35]. Consistence advocacy over the recent years had ensured the implementation of RoVs in countries which hitherto reported with a high rate of rotavirus-associated mortality in children. A country such as Nigeria, with the highest rotavirus mortality rate in Africa (≈55,000 deaths/annum) is yet to implement any of the acclaimed RoVs [36,37]. Sad still, more than 55 million children (≈41% of all children) lack access to RoVs globally [35]. By interpretation, ≈1 out of 2 children globally has access to RoVs. Besides, various reports have indicated lower efficacies of Rotarix^®^ and RotaTeq^®^ vaccines in sub-Saharan Africa and parts of Asia, which are the two regions of the world with the highest-burden of RVI, accounting for more than 90% of the total global infections [38,39].

Due to the high cost of RoVs, GAVI has continued to support most of the low-resource countries, especially in Africa and Asia, to enable vaccine implementation [40,41]. The recent effort of UNICEF in subsidising vaccine costs in low resource countries of Africa, Asia, the Caribbean, and South America is highly encouraging and commendable. In addition to this, UNICEF, through its strategic children’s health priorities, had worked so hard to secure the lowest possible children’s five-in-one vaccine price at an average of 0.84 cents/dose, which is almost half of the price giving to UN children’s agency [42]. This exertion reflects the work of UNICEF, on behalf of GAVI, towards affordable and sustainable vaccine supply for vulnerable children in developing countries. This consecutive three-year-purchase (2017 to 2019) pentavalent vaccine mixture is against diphtheria, tetanus, pertussis, hepatitis B, and *Haemophilus influenzae* type b infections. This is a relief on the financial burden incurred by the RoV implementation in such countries, and the continuity of such will progressively lighten the economic burden of childhood routine vaccines in these countries.

When Rotarix^®^ (RV1) and RotaTeq^®^ (RV5) were licensed in 2006, GSK offered its vaccine at USD 2.50 per dose (67% reduction over the lowest public price) to GAVI-eligible countries [43], while Merck & Co. Inc. offered it at USD 3.50 per dose (at ≥30 million purchased doses otherwise the price remains USD 5.0 per dose) [44]. This condition alone makes Rotarix^®^ more accessible because of its relative affordability and two doses prescription, unlike the RotaTeq^®^ with three doses prescription. Even at these prices, RoVs are still more expensive than most traditional EPI vaccines. In South Africa, for example, rotavirus and pneumococcal-conjugate vaccines were introduced in 2009. Being a middle-income country (MIC), it was not eligible for resource fund help from external bodies, and this has resulted into an over five-fold increment in the annual budget allocated to children immunisation alone [45]. In recent times, the two Indian WHO pre-qualified vaccines came up with relatively lower prices with Rotavac^®^ sold for about USD 1/dose and RotaSIIL^®^ sold for USD 2.5/dose [38].

The current award price/dose/product/supplier/calendar year for RoVs given to the 64 GAVI-eligible countries through UNICEF procurement are as follows: Rotarix^®^ (USD 2.10 per dose), RotaTeq^®^ (USD 3.2 per dose), Rotavac^®^ (USD 0.85 per dose), and RotaSIIL^®^ (USD 0.95 per dose) [37,46]. This current award price excluded Pan American Health Organisation (PAHO) countries, India, Vietnam, Indonesia, and fully self-financing countries. From the Centers for Disease Control and Prevention (CDC) vaccine price list, RV1 is USD 94.69 per dose while RV5 is USD 70.49, but from the private sector, RV1 is USD 120.95 while RV5 is USD 84.53 [47]. Compared to the UNICEF highest price data list for the Bacillus Calmette–Guérin (BCG) vaccine (USD 0.27 per dose), DTP vaccine (USD 0.19 per dose), DTP-HepB-Hib (Penta) vaccine (USD 1.2 per dose), measles vaccine (USD 0.4 per dose), OPV vaccine (USD 0.19 per dose), and yellow fever vaccine (YFV) (USD 1.44 per dose), RoVs are still very much expensive even at these subsidised prices, especially the RV1 and RV5.

Undoubtedly, RoVs are relatively more expensive than most other general childhood routine vaccines. This is one of the key reasons many LMICs could not implement these vaccines. The transition from GAVI-supported subsidiary to standalone financing is a crucial issue to those countries that are still benefiting from this subsidy. However, the justification for the health impact and cost-effectiveness of RoVs arises because of the cost implication of diarrhoea mortality and hospitalisation vis-à-vis the economic burden to the country. Eventually, after careful consideration of the policy and implementation surrounding the vaccines, countries are encouraged to introduce RoVs. After a thorough analysis of the diarrhoea and cost-of-illness against RoV effectiveness, efficacy, and the cost per disability-adjusted life year (DALY), RoVs were declared effective, beneficial, and life-saving in several countries that have implemented rotavirus vaccination into their national EPI in Africa [48,49], Asia [50], and Latin America and the Caribbean [51,52]. Forecasting data analysis had shown that if RoVs are implemented just in the GAVI-eligible countries only, over 600,000 deaths related to diarrhoea and rotavirus-related infections can be averted and save over USD 900 million. In another version of the forecast, peak vaccine “take” in the GAVI-eligible countries would have prevented 2.46 million childhood deaths and 83 million disability-adjusted life years (DALYs) from 2011 to 2030, with annual reductions of 180,000 childhood deaths [53,54].

Regarding formulation, packaging, and presentation, a model was designed to re-valuate the cost and cost-effectiveness of RoVs. In this model, Rotarix^®^ and RotaTeq^®^ are presented as a single-dose with about 5% wastage in terms of packaging. Rotavac^®^ is presented in a five-dose regimen resulting in smaller package volume but with a higher waste of about 25%, while RotaSIIL^®^ is presented as a two-dose regimen with a higher volume and about 5% wastage [55]. With this, Rotarix^®^ remained preferable due to its presentation and packaging. Moreover, the tendency of vaccine completeness among infants is higher in the Rotarix^®^ than any other licensed vaccines. In terms of their health impact, there was no significant preference as all of them provided highly competitive protection. Nevertheless, many analyses supported the continued use of Rotarix^®^, especially in GAVI-eligible countries.

All licensed RoVs are live-attenuated vaccines capable of reassorting with circulating field strains, which may lead to the emergence of more virulent strains [56]. Besides, poor seroconversion is currently observed in Africa and Asia due to transplacentally acquired IgG [25,57]. Albeit at a meagre rate, LAORoVs pose a risk of intussusception (IS) [38,58]. This risk was reported for the prototype human–animal reassortant live-attenuated oral vaccine, RotaShield^®^ from Wyeth (USA), which was withdrawn from the market in 1999 [59]. Post-marketing surveillance studies had shown an excess risk of 1–6 cases of IS at either the first or second dose for Rotarix^®^ and RotaTeq^®^ per 100,000 immunised children [38]. Similar rates have been reported for the recently licensed vaccines, Rotavac^®^ [60] and RotaSIIL^®^ [61]. In all cases, it was argued that the rate of intussusception is age-dependent [62,63].

Except for RotaSIIL^®^, all oral vaccines are to be maintained in a cold-chain to retain their efficacy and lifespan. In terms of production, the total capacity for RoVs is far below the global demand [33,64], which, therefore, calls for a more concerted effort for vaccine production and coverage. Apart from these associated challenges, the choice of RoV is constantly being restrained due to the financial status of the affected countries (the majority of which are LICs), government policies, and law-makers [65].

## 3. Initial Concepts to Produce Live-Attenuated Oral Rotavirus Vaccines

Early demonstration that an antibody in the animal lumen could offer protection against RVI and that natural multiple rotavirus infections in human can also induce heterotypic protection constituted an eyeopener than encouraged the development of LAORoVs [66,67]. In retrospect, many LAORoVs had been tried in neonates, infants, and toddlers but with mixed results, which probably retrogressed their further development. Animal rotavirus (ARV) strain, with the Jennerian approach, was first used as a potential RoV because of their antigenic similarities with the human rotavirus (HRV) subgroup 1. These strains are naturally attenuated during cross-species infections and offered cross-protection against HRV infections caused by subgroups 2 and 3 [68,69,70].

The bovine strain, RIT4237, was developed and cold-adapted from serotype G6 NCDV–Nebraska calf diarrheagenic virus [71]. Further, this Lincoln strain RIT4237, together with RIT4256, offered protections against HRV strain 2 in gnotobiotic calves [72]. One dose (1D) of this vaccine induced a seroconversion rate of 88% (detected by enzyme immunoassay, EIA) or 68% (detected by neutralisation assay) in children around 2 years of age [73], without any associated adverse events. Furthermore, this vaccine candidate proved to be non-reactogenic in neonates and babies between 6 and 12 months of age, with over 88% protection against diarrhoea in Finnish children when monitored for five months [74]. Variants of RIT4237 were tested in children from the USA [66,75], Gambia [76], and Rwanda [77] (Appendix A). These vaccine variants showed a high seroconversion rate in the developed countries more than the developing countries without any significant alteration from the infant feeding patterns, however, the vaccine is weakened by gastric acid, infant early exposure, and maternal antibodies.

When rhesus rotavirus monovalent (RRV-MV) and tetravalent (RRV-TV) reassortant vaccines were assessed in 3Ds with USA infants (4–26 weeks of age), with over two years of follow-up, the vaccines were well tolerated with no adverse events. The RRV-MV produced 40% efficacy against rotavirus diarrhoea (RVD), 73% against very severe rotavirus gastroenteritis (VSRVGE), and a 67% reduction in emergency medical visits, while RRV-TV respectively produced 57%, 82%, and 78% [78]. Both vaccines offered protection against serotype 1 rotavirus, but only RRV-TV offered protection against non-serotype 1 over the second year [78]. Rennels and co-workers reported a similar significant reduction in the incidence of RVGE by the two vaccines in USA children between 5 and 25 weeks of age, with RRV-TV having a more protective effect than RRV-MV. RRV-TV prevented 49% of cases of rotavirus episodes, 80% of cases of VSRVGE episodes, 100% of cases of dehydrating RVD, and an 82% reduction in all cases of dehydrating diarrhoea [79]. Variants of RRV were tested in children from Venezuela [80], Sweden [81], and Finland [82,83] (Appendix A). There were mixed results in all these trials. Generally, the vaccine was reactogenic and highly shed in older children or with increasing dose. The vaccine variants induced homotypic and heterotypic protections better than BRV vaccine candidates, and the heterotypic protection increased with age. The immunogenicity and safety appeared to be enhanced by the pre-existing neutralising antibody, and the efficacy increased with clinical severity. The infectivity and immunogenicity increased with dose, but the seroresponse decreased with age. Lastly, the vaccines were moderately affected by the breastfeeding and maternal antibody.

In addition, Vesikari and co-workers compared the immunogenicity, viral shedding, and reactogenicity of RRV-1 and RIT4237 in 6–8 months Finnish infants. RRV-1, given at 1.0 mL (≈10^8.3^ PFU/mL) from a 1:10 vaccine dilution had a history of two times passage in the cynomolgus monkey primary kidney epithelial cells (CMPKEC), seven times passage in AGMKC, and seven times passage in the diploid foetal rhesus monkey lung DBS-FRhL-2 cells [84]. This vaccine candidate produced ≈ 10^6^ PFU/mL in MA104 cells [85] and was stored at −70 °C. Similarly, RIT4237 (vaccine lot L1109), given at 0.5 mL/dose, was passaged 147 times in the FBKC with final production at the 154th passage in the AGMKC at 10^8.3^ TCID_50_/dose, lyophilised, stored at −20 °C, and dispatched by Smith Kline RIT, Rixensart, Belgium [72,86]. Antibody response to vaccine infection was 88% in the RRV-1 and 75% in the RIT4237. Further, 84% of viral shedding was observed in infants that received RRV-1, while 21% was observed in the RIT4237 recipients within a week after vaccination. More significantly, unlike RIT4237, RRV-1 was associated with high fever (>38 °C) between three and four days post-vaccination in 64% of infants and watery stools 4–5 days post-vaccination in 20% of infants [87]. Therefore, RRV-1 (MMU18006) oral RoV was more immunogenic than bovine RIT4237 in infants, but it was associated with very high adverse events.

An early observation of the similar antigenic epitope of bovine and human rotavirus neutralising antigens has spurred the investigation into the use of attenuated animal rotavirus strains as an oral vaccine candidate [88,89]. Besides, an animal strain vaccine candidate offered cross-protection in animal studies against animal and human rotavirus infections [70,90]. Highly attenuated Lincoln NCDV (serially passaged for 37 times in FBKC) offered protection against subsequent infection by Cody NCDV in tested calves but induced an insufficient neutralising antibody in volunteered adults [71]. The attenuated WC3 BRV vaccine candidate induced less viral shedding with more homotypic protection in infants but the protection against more than one serotypes increased with age [91,92], and it appeared to be boosted by the pre-existing naturally acquired protection [91,93]. However, this vaccine candidate performed poorly with <48% protection in the LICs against severe rotavirus diarrhoea (SRVD) [94], and the follow-up protection over two years was very poor (Appendix A).

The use of the HRV strain as a LAORoV started with the common neonatal strains G1–G4, which had the VP4 gene conserved and naturally attenuated [66]. Such strains include serotype 1 (M37), serotype 2 (1076), serotype 3 (McN 13), and serotype 4 (ST3). The HRV strain M37 candidate was a serotype G1P2A nursery strain isolated from asymptomatic 2-day old neonate from Venezuela and passaged 29–30 times. The vaccine was well tolerated at 10^4^ and 10^5^ PFU/mL as tested in 102 and 39 infants, respectively, but was characterised with fever within the first seven days. Anti-rotavirus IgA seroresponse was 47% in 10^4^ PFU/mL and 76% in 10^5^ PFU/mL [95]. This, therefore, means that the higher dose offered more protection with a higher immunogenic response, but the efficacy of this dose remained unconfirmed [95]. However, the response seemed to be serotype-specific, yet two doses of vaccine titre at 10^4^ or 10^5^ PFU/mL within 1.0–2.5 months increased the overall serological response rate in USA infants <6 months of age to 88% [96]. Furthermore, 70% of infants that received 10^5^ PFU/mL showed neutralising seroresponses, but surprisingly, 70% also shed the virus [96]. Thus, the high rate of vaccine shedding may likely be a problem with this vaccine candidate because this might promote viral transmission. From the reactogenic and antigenic studies, 50% of Venezuelan infants (10–20 weeks of age) that received M37 at 10^4^ PFU/mL showed a serum rotavirus IgA response, 64% showed a neutralising antibody response to the M37 strain, 27% to the human serotype 1 Wa strain, and 27% to the human serotype 4 neonatal ST3 strain [97]. This shows that the neutralising antibody response is predominantly vaccine strain-specific, as the highest response was found against M37 [96].

The RV3 (G3P2A) vaccine candidate was an age-dependent vaccine tolerance, which induced sustainable heterotypic protection [66] without any side effects or shedding [98], but the homotypic infections failed to confer immunity against rotavirus re-infection [99]. In Australia, heterotypic protection offered by this vaccine candidate appeared to decrease in the second year of follow-up and was also found to be mitigated by the maternal antibody [99,100]. Variants of IGV-80-3 HRV, cold-adapted (*ca*) and temperature-sensitive (*ts*) vaccine candidates, have been tried in animal [101] and human [102]. This was passaged from 37 to 25 °C with remarkable immunogenicity in piglets and mice [103] (p. 315). It contains the serotype 1 and subgroup II conserved antigens—VP4 and VP7 [101,104]. Examples include strain D (containing VP4: 1A; VP7: 1), DS-1 (containing VP4: 1B; VP7: 2), and human–human rotavirus reassortants like Wa x DS-1 (containing VP4: 1A; VP7: 2) and Wa × P (containing VP4: 1A; VP7: 3) [101]. The details of the efficacy and safety trials of other HRV vaccine variants are described in Appendix A.

Reassortant vaccine candidates were also produced to improve the LAORoV efficacy against RVI. These were made with either human–rhesus or human–bovine rotavirus strains based on the circulating human strains [105,106]. The popular reassortants are the BRV UK strain serotype 6, RRV MMU18006 serotype 3, and HRV strains (serotypes 1, 2, 3, and 4). These heterologous vaccine candidates generally come as a single reassortant with human strain major neutralising VP7 and to some extent, VP4 [107]. With this, reassortant vaccines are believed to be more effective and offer broader heterotypic protection because of the genetic variability offered by the neutralising VP7 and VP4. Human VP7 serotype G1 D × RRV at 10^4^ PFU/mL and VP7 serotype G2 DS1 × RRV at 10^5^ PFU/mL were tried in Finland. The vaccine showed similar reactogenic and adverse events with the parent strains, exhibited homotypic protection, but asymptomatic protections reduced drastically over two years of follow-up [108]. Heterotypic protection offered by the higher dose failed to protect Peruvian children [66]. In Venezuela, similar reactions and protections were observed with high vaccine “take” and shedding [109]. A quadrivalent reassortant vaccine–RRV serotype 3 VP7 and three human–RRV reassortant strains (D × RRV (serotype 1 [VP7]), DS1 × RRV (serotype 2 [VP7]), and ST3 × RRV (serotype 4 [VP7]), showed similar reactions and dose-dependent protections in Venezuela [110], however, vaccine interference was suspected because individual doses displayed higher seroresponse than the combined doses (Appendix A).

The presence of strain interference in the human–rhesus rotavirus reassortant polyvalent vaccine candidates has been speculated [110]. A careful comparison of the reactogenic and immunogenic properties of RRV-DV [109] and RRV-TV [110] showed no significant difference in terms of reactogenic features, vaccine titre, homotypic protection, and vaccine shedding. This observation seemed contrary to the strain interference speculated. Rather, a synergistic protective effect may be offered by these polyvalent reassortant rotavirus vaccine candidates. Nevertheless, this heterologous vaccine candidate displayed some levels of heterotypic protection against all the four common HRV strains, which are considered epidemiologically significant. The available report showed 58% overall protection in USA infants and 32% protection in Peruvian infants at a 4 × 10^4^ PFU/mL vaccine dose [66]. One might consider that increasing the dosage of this vaccine will improve the seroconversion rate; however, this will rather complicate the adverse events, especially in immunodeficient children [110,111]. Focusing on the vaccine preparation in such a way that each component of the mixture is maximally “take” will improve the seroconversion and immunogenicity of this vaccine candidate.

With the apprehensive reactogenic effects of RRV in human, the reassortant of HRV serotypes 1, 2, 3, and 4 bearing gene 9 (encoding major neutralising antigen VP7) with BRV strains WC3 and UK may be a better alternative. The bivalent reassortant W179-9, carrying the antigenic phenotype of human serotype 1 VP7 antigen and bovine serotype 6 backbone (including the VP4 antigen) administered at the highest titre of 10^7.5^ PFU/mL per dose induced no adverse event in 2–11 months Pennsylvanian children [112]. Breastfeeding did not affect this reassortant vaccine. Serotypes 1 and 6 specific serum neutralising antibodies were equally induced; however, pre-existing seropositive infants failed to seroconvert to the vaccine, especially the serotype 1. A booster dose of this reassortant vaccine seemed to be effective to overcome the pre-existing immunity in the seropositive aged children. Pre-existing seropositive in infants, 5–11 months of age, appeared to inhibit the serum neutralising antibody against the major serotype antigens present in the W179-9. This was unlikely in the 2–4 months infants where the heterotypic immune response was observed with a booster dose, even against the serotype 3 strain SA11 [112].

## 4. The Renaissance of Live-Attenuated Oral Rotavirus Vaccines Today–The Journal so Far

Rotarix^®^, RotaTeq^®^, Rotavac^®^, and RotaSIIL^®^ are available internationally with WHO prequalification [14]. By summarising the historical background of these vaccines, the first human rotavirus vaccine candidate was designed with the monovalent Nebraska calf diarrhoea virus (NCDV) BRV-RIT4237 strain G6P6[1] [73] based on the “Jennerian concept” [113]. However, this vaccine was withdrawn due to inconsistent results in the clinical trials [114,115]. A simian rotavirus vaccine called rhesus rotavirus (RRV-MMU) was developed, but its protection was similar to that of BRV-RIT4237—inconsistent and highly discrepant clinical data. Subsequently, RRV was reassorted with genome segments encoding VP7 representing human G genotypes (G1, G2, and G4) to form the tetravalent RotaShield^®^ vaccine produced by Wyeth-Lederle Vaccines (USA). RotaShield^®^ was tested in a large, placebo-controlled trial with an efficacy of more than 70% [116].

However, at post-licensing, the vaccine was withdrawn due to an increased rate of IS [117]. A second bovine rotavirus vaccine candidate named WC3 was developed initially using the G6P[5] strain. This showed heterotypic protection [92], but subsequent trials provided no significant protection [118]. The WC3 was later reassorted with the human rotavirus VP4 and VP7 encoding genome segments to produce the pentavalent RotaTeq^®^ vaccine. RotaTeq^®^ was produced by Merck and was considered safe because of its consistent protection in clinical trials [119,120]. The monovalent vaccine G1P1A [8] serotype named Rotarix^®^ was also developed based on the cross-protection from multiple natural infections [121,122]. Consequently, RotaTeq^®^ and Rotarix^®^ have been licensed since 2006 [123,124], followed by Rotavac^®^ and RotaSIIL^®^, Indian made LAORoVs, which were also licensed for global used. Rotavin-M1^®^ made in Vietnam, and LLR-37 from China remain licensed to be used regionally. Other notable vaccines in phase trials are RV3-BB infant vaccine (prepared by Murdoch Children’s Research Institute, Melbourne, Victoria in Australia and Biofarma, Bandung in Indonesia—going to phase III trial), BRV-TV (prepared by Shantha Biotechnics, Hyderabad, India—Phase III trial), and BRV-PV (prepared by Instituto Butantan, Butantã, São Paulo, Brazil—Phase I trial) [38,125] (Table 1).

## 5. Comparative Analysis of Vaccine Coverage, Effectiveness, and Efficacy

Available reports have shown considerable data analysis in the pre-clinical trials of Rotarix^®^ and RotaTeq^®,^ and it can be concluded that vaccine efficacy differs by regions and countries. Countries with high child mortality rates due to RVI usually show relatively lower efficacy [126]. The efficacy and effectiveness for Rotarix^®^ and RotaTeq^®^ were quite high in HMICs ranging from 85% to 98% [127,128], though with differential genotype-specific protection [129]. However, average performances were observed in the LICs in Africa and Asia, with efficacies ranging from 51% to 64% [130,131].

Currently, more than 74 countries (41 from GAVI-eligible countries) exclusively adopt a two-dose regime for Rotarix^®^, which are usually given at 2 and 4 months after birth, while 5 GAVI-eligible countries adopt a three-dose regime for RotaTeq^®^ at 2, 4, and 6 months with 9 countries practising both regimens. Generally, the first dose of RoV should have been administered before 2 months, and all doses should have been completed before 8 months as LAORoV administration is highly discouraged for infants less than 1 month or children after 8 months. This is to minimise the incidence of IS, which usually occurs within the 7 days after the first/second dose or 21 days after the second dose [132,133].

There was a general reduction in the vaccine effectiveness (VE), especially against heterotypic infections, as observed in the LICs [134,135] (Table 2). About 10% of lower vaccine efficacies have been observed in Latin America as compared with the USA and European countries [121,127]. Much-reduced efficacies were observed in Africa and Asia, especially in the second year of follow-up [136,137]. Though Rotavac^®^ and RotaSIIL^®^ were in a few years back licensed for worldwide usage, the prior pre-clinical analysis showed similar vaccine efficacy to that of Rotarix^®^ and RotaTeq^®^ in the LICs [60,61] (Table 2). There is convincing evidence that the introduction of RoVs into more than half of Africa had greatly reduced the proportion of diarrhoea-related hospitalisations and cases of SGE in infants due to RVI [138,139].

From the available resources regarding vaccine effectiveness, coverage, and the impact of post-introduction vaccine surveillance (Appendix A), it can be deduced that the significant impact of the vaccines correlated with the percentage coverage in the representative countries of the Americas and Europe [129,140], however, the opposite is the case for Africa and Asia. With an average upper limit coverage in Germany and Moldova, for example, over 80% vaccine efficacy was achieved within two years. However, over six years of surveillance studies of VE in Ghana with extremely high coverage limits (93–100%) produced 55.5% VE against the SRVGE. Rwanda and South Africa had an extremely high percentage coverage, achieving 75% and 76.9%, respectively, within two years; even so, a much lower VE was achieved in Malawi, Botswana, and Bangladesh despite their high coverage indices (Appendix A). In spite of high vaccine coverage in Africa and Southeast Asia, the overall vaccine efficacy against all forms of rotavirus gastroenteritis remains between 54% and 76% (Table 2). It is very important, therefore, to re-emphasise that with these average performances of the vaccines in Africa and Asia, a greater proportion of the rotavirus-related GE and hospitalisation has been averted, which is of significant perspective in the public health at large. From the WHO/UNICEF Estimates of National Immunization Coverage (WUENIC) perspective, among the 98 countries that introduced rotavirus vaccines, 38 countries have an extremely very high vaccine coverage of between 90% and 100%, 23 countries have 80–89%, 17 countries have 70–79%, 4 countries have 60–69%, and 16 countries with <60% coverage; however, official country reports showed 35 with extremely very high vaccine coverage, 26 with extremely high vaccine coverage, 7 with very high vaccine coverage, 6 with high vaccine coverage, and 9 with average vaccine coverage, respectively [141].

Furthermore, apart from the RoVs saving the lives of infants and children, it also reduces the burden of healthcare organisations. Analysis by Aliabadi and co-workers from the Global Rotavirus Surveillance Network (GRSN) perspective purportedly showed that a nearly 40% reduction in the number of AGE hospitalisation as a result of RVI in children <5 years has been achieved between 2008 and 2016 in the WHO regions [142,143]. Interestingly, reports from the USA have attributed the reduction observed in all GE and rotavirus hospitalisations in children from 5 to 17 years as well as adults from 18 to 64 years of age to RoV benefits [36].

RoVs also offer translational protection. As high as 26% herd immunity for non-vaccinated children has been reported [144]. This is a highly beneficial effect on wider vaccine coverage. This effect was first observed in the USA among the ineligible unvaccinated babies (>6 months) [145], and as high as 50% protection was also observed in adults [146]. Subsequently, this public health impact of RoVs has been reported in other HMICs [147,148], but this indirect protection (herd immunity) seems to be lacking in the LICs [131]. To encourage herd immunity offered by the RoVs in such a high epidemic region, consistent immunisation and wider vaccine coverage are inevitable.

However, vaccination is constantly being refrained in certain regions of the world for unknown and unsure reasons; therefore, a need exists for re-sounding advocacy for vaccination. Human health is constantly under attack from environmental pollution, unhygienic practice, poor nutrition, human–wild animal interactions, resistance pathogens, and the sporadic evolution of untyped pathogens. This is the reason infants are given more vaccines nowadays against a wider range of infections. Relenting the wider vaccine coverage can eventually lead to the emergence of old and resolved disease conditions with the possibility of a more virulence pathogen/mutant. This invariably will overburden the public health system.

Regarding RoVs, WHO has set a target of 90% rotavirus vaccine coverage nationally and at least 80% in every sector/district. In recent publications, the HICs only achieved an average of 45% while the self-financing MICs managed to achieve 22%. Furthermore, the GAVI-eligible countries achieved 28%, and the global rotavirus vaccine coverage climbed to 28% from the formal 20% [149,150]. In another publication by WHO, 101 countries were stipulated to have introduced RoVs with about 35% global coverage [33].

WHO had reported that 49 from 98 countries that introduced RoVs were supported by GAVI and 8 from 10 countries that applied for GAVI assistance have just been granted an approval [37]. GAVI has continued to help LMICs to introduce RoVs into their necessary childhood routine vaccines through the UNICEF initiative procurement with 91% market demands for RV1 over RV5 purposely because RV1 is cheaper and administered in two doses [37,151]. With the incoming vaccine implementations, more is expected from GSK, which implies technically less supply from Merck and Co. However, Merck has continued to improve in its supply of RoV products to GAVI-eligible countries [152]. It is very important, therefore, to know that in the last two years, GAVI market supply had constantly been faced with inadequately supplied vaccine products from GSK due to technical difficulties with reducing market supply from Merck. This posed a challenge for the 2018 and 2019 vaccine roll out to the extent that the affected countries unanimously planned to switch to the recently prequalified Indian RoVs [64]. This condition arrived because of the GAVI-eligible graduating self-financing countries that may not procure their vaccines through UNICEF vis-à-vis more implementing countries with increasing infant populations [153].

These unilateral heavy market demands from GSK, though they may affect the effective production and global supply in the long run as well as impose vaccine pressure on the strain selectivity, will directly enhance wider coverage and probably contribute to herd protection, especially in the LICs of Africa and Asia. Apart from the wider coverage, valences of the vaccines also have a significant effect. RotaTeq^®^, being a pentavalent vaccine, is likely to produce higher herd immunity as compared with the Rotarix^®^; however, the majority of the available reports showed striking similarities between the two vaccines against all severity of RVIs leading to hospitalisation and against specific circulating/predominant strains in Europe [185], North America [186], Central America [187], South America [188], Asia [57], and Africa [180].

Generally, there has been an 11% increment in the RoV coverage in the GAVI supported countries from 2017 to 2018, with a total of 39%. Africa has continued to be the leading continent in the implementation of RoVs with 39 from 54 (72%) geographical countries introducing RoVs into their EPI [36]. Sadly, with all the implementation and the vaccine roll-outs, Africa still accounts for 49% worldwide death of children due to rotavirus-related infection, Asia 33%, and the rest of the world 18%. Alarming still, the last report on the average mortality rate per day from RVI in Africa showed more than 330 children [34]. This is hindsight to the observation of Armah and co-workers on the efficacy of pentavalent RoV against SRVGE in infants from developing countries of sub-Saharan Africa [136]. Unfortunately, Africa and Asia are currently the two continents with the highest burden of RVI [142]. This, therefore, means that concerted efforts are still needed in these regions to curb RVIs.

Strategically, birth dose/booster dose of RoVs, non-interference rotavirus vaccination, parenteral administration of subunit RoVs, and introduction of non-replicating rotavirus virus-like particles (RV-VLPs) are an available concerted paradigm against the residual burden of RVIs [188,189,190]. As much as these factors could enhance the overall vaccine efficacy and effectiveness against all forms of RVGE, cautions are needed to understudy each of these factors extensively before implementing them. For example, waning immunity after a year of vaccination may likely predispose children in developing countries to rotavirus diarrhoea. In Malawi, there was no significant improvement in VE when two vaccine doses were compared with three vaccine doses with RV1 against the all-cause of GE [179]. Further, a reduced VE was observed in the second year of follow-up after RV1 vaccination [179]; this may probably be due to prevalent waning immunity in Africa. However, the opposite is the case for South Africa, where 2- and 3Ds of RV1 at 6, 10, and 14 weeks were compared during the two consecutive rotavirus seasons [191]. There was a general increase in VE from 2Ds to 3Ds [192].

Furthermore, a meta-analysis by Burnett and co-workers showed that an additional booster dose of RoV with the measles vaccine at 9 or 12 months could avert as much as 29,400 additional rotavirus deaths if the VE is boosted by 50% of the difference between the first year and the second year VE [189]. Notwithstanding, the seasonal occurrence of RVIs, vaccine coverage, age distribution, strain divergence, co-infections, general nutritional status, cold-chain storage facility and transportation, and cost-effectiveness are some of the germane factors that could negate the regional implementation, efficacy, and effectiveness of RoVs. These factors are further discussed below.

## 6. Factors Affecting Rotavirus Vaccine Efficiency

Poor rotavirus vaccine absorption in the gut has been predicted as a major underlying factor affecting vaccine efficacy, particularly in Africa and Southeast Asia [193,194]. Any intricate factor imposed on the “take” of LAORoVs in the gut may not necessarily prevent viral replication and subsequently viral shedding. This, therefore, means host or environmental factors that enhance viral shedding and expulsion from the gut may probably limit the efficacy of RoVs and the mucosal immune response to RVI. Some of these factors are intrinsically developed in the host, acquired at birth, or imposed by the environment. However, it is technically impossible to generalise these factors as each differs in their significant effect. Further, each of these effects seems to be regionalised.

### 6.1. Breastfeeding

Breastfeeding has been argued as a factor against RoV efficacy, but the evidence seemed inconclusive. The breast milk IgA neutralising activity together with other non-antibody milk proteins such as lactoferrin, lactadherin, mucin, and butyrophilin have been shown to inhibit the replication of animal and human rotaviruses in intestinal epithelial cells [195] and confer protection against symptomatic RVI in infants [196,197].

A report was put forward to show that higher titre breast milk IgA in Indian nursing mothers was able to neutralise and inhibit the infectivity of LAORoV more than American nursing mothers [198]. The neutralising activity was higher in the human-derived RoVs (RV1 and 116E) than the human–bovine reassorted vaccine (RV5) and sequentially decreased from Indian nursing mothers to Korean, Vietnamese, and American [198]. Furthermore, a negative correlation has been observed between the vaccine immunogenicity and breast milk antibody/non-antibody levels in Indian and South African nursing mothers when Rotarix^®^ and Rotavac^®^ were assessed in them. However, this correlation was not found in American nursing mothers [199]. In a randomised trial with Rotarix^®^, abstention from infant breastfeeding for at least 1 h before and after each vaccine dose had no significant effect on the rotavirus immune response by the infants [200]. Likewise, the serum anti-VP6 IgA seroconversion ability with 30 min of breast milk withheld before and after each of the vaccine dose had no impact on the immune response [201,202,203]. Gastañaduy and co-workers reported similar VE between the breastfed (50% VE) and mixed-fed (51% VE) babies that received RV1 FDs against RVI with hospitalisation in Botswana [204].

Similarly, one other report from Africa showed that the efficacy of RotaTeq^®^ against any RVGE severity caused by G1–G4 in infants never breastfed, sometimes breastfed, and exclusively breastfed was 68.3%, 82.2%, and 68.0%, respectively, and the efficacy against SRVGE in the three groups was 100%, 95.4%, and 100%, respectively. This means that breastfeeding has no significant adverse impact on the efficacy of RV5 [205]. This result was quite similar to the early assessment of RRV-PV, where similar seroresponses and protection in breastfed and non-breastfed children were observed in USA infants [206]. However, a meta-analysis previously carried out on the effect of breastfeeding on oral RRV in infants 2–5 months showed a significant adverse effect on seroconversion (48% in breastfed vs. 70% in bottle-fed babies) [207]. More still, Dennehy and co-workers reported 10–12% reduced vaccine “take” in exclusive breast-feeding and mixed-feeding as compared with the infant formula children [168] while assessing the safety and immunogenicity of RV1.

It seems RV1 may be vulnerable to breast milk neutralising IgA and other non-immunogenic proteins, perhaps because it is a homotypic vaccine strain. However, increasing the vaccine dose has been suggested to overcome the effects of breast milk and interference from other routine childhood vaccines [79]. One classic example is the work of Ali and co-workers from Pakistan, where 2Ds of RV1 produced a 16.6% rate of anti-rotavirus IgA seroconversion in infants with a 1-h delay of breastfeeding and 29.1% seroconversion in unrestricted breastfeeding infants; however, in the 3Ds regimen, these increased to 28.2% and 37.8%, respectively [201]. Apart from the marginal increase in the rate of seroconversion in unrestricted breastfeeding infants, the report might have indicated the advantage of increased vaccine doses against breastfeeding.

### 6.2. Maternal Antibodies Acquired Transplacentally

Rotavirus-specific IgG obtained transplacentally has been shown to reduce the rate of seroconversion to RoVs [26,208]. Infants acquired maternal IgG during birth because it is the only immunoglobulin that can cross the placental and this natural event prepared the infants for the first encountered infections, including RVI. There was a strong correlation between the presence of rotavirus-specific antibodies in the maternal serum/colostrum and the infants’ cord serum and milk. This hypothesis has been verified in many parts of the world before the advent of RoVs [209,210]. Some of these specific antibodies include secretory immunoglobulin, IgG, IgA, and IgM. These antibodies lower the viral replication [211] and might be responsible for the lower vaccine “take” in some regions. Rotavirus aggregation has been detected in the stool of breast-fed infants, which indicates antibody-induced passive protection by IgA and IgG [212]. Likewise, serotype-specific antibodies have been observed from the colostrum, breast milk, and cord blood [213]. In India, recuperating rotavirus-positive and hospitalised-infants showed higher homologous neutralising antibodies to circulating serotypes G1, G2, G4, and G9 than their respective mothers. However, the mothers showed higher neutralising antibodies to non-circulating and animal rotavirus strains, simian G3 and bovine G6, G10, and this invariably predisposed the infants to these strains perhaps if these strains become prevalent [214]. For this reason, serotype diversity and prevalence may be critical to the vaccine effectiveness and efficacy in LICs and LMICs.

### 6.3. Microbiota/Probiotic Diversities

The quality of microbiota/probiotic diversities in children from different resource countries have been comprehensively compared [215,216] in terms of how it affects the immune system against gut infections [217,218]. This has been considered critical to immunity against RVI, based on animal studies [29,219]. The presence of gut commensals such as *Lactobacillus rhamnosus GG* (LGG), *L. acidophilus*, *L. reuteri*, and *Bifidobacterium lactis Bb12* decreases the clinical symptoms of RVI [220,221] and improves the mucosal B-cell response to the rotavirus challenge in animal studies [222,223]. A significant correlation between the composition of the infant gut microbiome and response to rotavirus vaccination has been observed in Ghana [28]. Here, *Streptococcus bovis* was highly abundant in the RoV responders (infants producing IgA ≥20 IU/mL) more than the non-responders (infants having their IgA ≤20 IU/mL), and this microbiome composition was very similar to healthy Dutch children [28].

Similarly, the comparison of microbiota among the responsive vaccinated children producing IgA ≥20 IU/mL from Pakistan and healthy unvaccinated children from the Netherlands showed the presence of a comparably high level of *Clostridia* and *Proteobacteria* such as *Serratia* and *Escherichia coli*. This further showed how these probiotic bacteria enhanced anti-rotavirus IgA seroconversion in the responsive vaccinated children who hitherto had serum IgA <20 IU/mL [29]. Coincidentally, bacteria, which produce toxigenic lipopolysaccharides (LPS) can act as an adjuvant with the RoVs to induce an immune response against rotavirus and enteric infections. The lack of effective LPS-secreting bacteria in the children of developing countries may be one of the reasons for lower vaccine “take” and hence poor efficacy. Pragmatically, enhancers of intestinal secretomes and mucosal immune response such as probiotics will improve the immune response to LAORoVs because these vaccine candidates replicate in the gut.

In India, an abundance of pre-vaccination bacterial taxa in infants’ stool vaccinated with RV1 showed a modest correlation to rotavirus shedding after the first dose. Parker and co-workers discovered in Vellore, India, that though vaccine responders tend to harbour more enteropathogenic bacteria than the non-responders, increasing the vaccine dose is not tantamount to increased microbiota activities [224]. In essence, 26% responders vs. 13% non-responders were observed for 1D of RVI and 24% responders vs. 23% non-responders to 2Ds of RV1 [224]. This means there was no strong correlation between the intestinal microbiota at the time of vaccination and RV1 immunogenicity [224]; however, rotavirus shedding was found associated slightly more with pre-vaccination bacterial taxa in the gut.

There was no concrete evidence of the probiotic bacteria promoting the viral replication and mucosal immune response in human as found in the animal models, perhaps because animal feeds are more of roughages than human. In the same Vellore in India, Lazarus and co-workers discovered a modest relationship between the preponderant LGG in infants’ stools at the time of vaccination and the increased rate of rotavirus shedding after the first dose of RV1 [225], however, this outcome left them with various unanswered questions. Despite the 10^10^ daily supply of probiotics for 6 weeks, the level of LGG remained low (<1%) compared with other intestinal bacteria and had no significant effect on the overall intestinal bacteria diversity [224,225].

Hitherto, about a 58% seroconversion to RV1 had been observed in Vellore, India [226]. From the same place, Lazarus and co-workers evaluated the effect of probiotic (10^10^ probiotic LGG/day) and zinc (5 mg/day) supplementations for seven weeks on the RV1 immunogenicity in infants co-administered with OPV. The total seroconversion observed after 2Ds of RV1 at 6 and 10 weeks was 39.4% (probiotic + Zn), 30.9% (probiotic), 28% (Zn), and 27.4% (vaccine minus supplementation). Furthermore, the pre- and post-vaccination IgA level ≥ 20 IU/mL respectively showed 25.5% vs. 51.8%, 32.4% vs. 55.2%, 29.4% vs. 49.7%, and 34.1% vs. 51.1% in infants for each of the classified supplements with either PPP or ITT analysis [225]. A similar trend was observed for the GMT values at the pre- and post-vaccination periods. This suggested that these supplementations seemed not to augment the vaccine efficacy. The authors further stated that seroconversion seemed to correlate with viral shedding. After the first dose of RV1, 14.4% of the infants shed the virus on day four, 17.3% on day seven, and 23.6% between four and seven days, corresponding to 68.7% seroconversion in all the vaccine responders/shedders and 45.2% in the non-responders/non-shedders [225].

In this evaluation, there were no significant adverse events caused by this intervention or vaccine and all the associated events were fully recovered [225]. Co-administration with the OPV and maternal anti-rotavirus antibody effect against the vaccine efficacy may not be ruled out completely in these supplement assessments. However, these effects have been proved inconsequential to some extent. Infants receiving zinc or probiotic supplementations, therefore, may not display significant improvement to the low immunogenicity of the rotavirus vaccine in the low resource urban community of India. Further investigations into the effect of probiotics on the immune response to oral vaccines are urgently needed. The a priori of this is that complex clusters of microbiota, micro-supplements, and vitamins may be needed for the concerted stimulation of the mucosal immune response to LAORoV.

### 6.4. Malnutrition

General malnutrition caused by the lack of vitamins A and D and poor serum zinc levels is common in sub-Saharan Africa and Asia. Poor nutritional status contributes to the lower efficacy of RoVs in these regions [227] because of the associated innate and adaptive dysfunctions [228]. Usually, malnutrition is manifested as a poor weight for age, height for age, and weight for height. One classical report from Botswana showed 75% VE of RV1 against RVGE in well-nourished as compared with the malnourished children with -28% VE [212]. Similarly, Bar-Zeev and co-workers from Malawi purportedly reported a drastic reduction in the RV1 effectiveness from 78.1% in the well-nourished babies against 27.8% in the malnourished babies [137] as well as in Kenya, where the effect of malnutrition drastically affected the VE [229].

With RV1 in Zimbabwe, 6–11 months old infants with normal height for age showed VE of 71% with FD vs. 71% with AD, but infants with stunted height for age showed 45% with FD vs. 37% with AD. Again, ≥12 months old children with normal height for age had −35% VE with FD vs. −34% with AD compared with the children with stunted height for age who had −67% with FD vs. −62% with AD [230]. The technical interpretation of this was that malnutrition and waning immunity interplayed to mitigate the vaccine effect. Furthermore, Linhares and co-workers reported the effect of malnutrition against the effectiveness of RV5 in Brazil. It was concluded that malnourishment might interfere with the rotavirus vaccine effectiveness (RoVE) in the developing countries [231]. Contrarily, Perez-Schael and co-workers statistically stated that RV1 protected the well- and mal-nourished infants equally (74.1% vs. 73.0% VE against SRVGE and 60.9% vs. 61.2% VE against any SRVGE) [232]. It seems the available data regarding the effectiveness of rotavirus vaccines in the mal-nourished is debatable, and increased sample sizes are currently the suggestion of observers [233]. Notwithstanding, the role of nutritional status cannot be ruled out in the general state of wellbeing and in boosting the immune response against infections.

### 6.5. Co-Infection

Studies have shown the negative impacts of co-infections at the time of rotavirus vaccination. Such infections include enterovirus infection [234] and bacterial enteric pathogens [235,236]. Beyond this, co-infections with rotavirus generally aggravate the clinical symptoms and prolong hospitalisation [237,238]. Paradoxically, the RoVE seems to be positively correlated with the severity of the infection. Patel and co-workers reported a 58% reduction in severe (Vesikari score ≥11) rotavirus diarrhoea and a 77% reduction in very severe (Vesikari score ≥15) rotavirus diarrhoea infection [239]. Similarly, Pringle and co-workers from Bolivia observed 54% VE and 72% VE against severe and very severe rotavirus infections, respectively [240]. Furthermore, VE increased from 53% to 70% against severe and very severe rotavirus infection in Botswana [204]. However, the severity of infection is expected to be exclusively caused by rotavirus. One typical example of extremely low VE against non-rotavirus AGE is found in the work of Boom and co-workers, where 15% VE was observed against non-rotavirus AGE while looking at the sustainable protection of RV5 during the second year of life in USA paediatric hospitals [241].

### 6.6. Overage

Mitigated RoVE/efficacy is now being reported in children ineligible to rotavirus vaccination due to overage. Correia and co-workers reported a decline in VE in vaccinated children that were > 12 months old [242]. Lower VE for children out of the acceptable age range is very common in the LMICs as compared with the HICs. The report from Nicaragua showed a significant reduction in the risk of RVI in vaccinated infants between 8 and 11 months as compared with the children between 12–19 months [239], the same also from Moldova [243]. Apart from the likelihood of IS in the vaccinated aged children, the occurrence of lower VE in them is not substantial enough to discourage vaccination in them, after all, there is a chance of herd protection, which can spill over from the vaccinated infants to the unvaccinated children. A few numbers of the countries with information on the VE in <12 and ≥12 months old children have been reported in Europe [185,244], North America [119,245], Central America [178,246], South America [240,247], Asia [131,148], Middle East [248], and Africa [230,249] (Appendix A). Both RV1 and RV5 produced almost the same efficacies in the two age groups from the HIC; however, the majority of the significant differences are observed in the moderate to high rotavirus mortality rate countries from the MICs and LICs, where the VE reduced with age.

The proposition that RoVE may decrease with age seems debatable. All rotavirus immunisation should have been completed by 32 weeks of age to minimise the occurrence of IS [250]. This recommendation may not go well with the LICs and MICs, where access to health facilities and vaccination is adversely extenuated. A report from Patel and co-workers published in 2012 assessed the benefit and risk in age-restricted vs. unrestricted rotavirus vaccinations. The outcome of this model analysis showed that in LICs and LMICs, restricting the first dose of RoV to < 14 weeks would prevent 155,800 rotavirus deaths and cause 253 potential IS, while an unrestricted vaccine schedule would have prevented 203,000 and potentially cause 547 IS [251]. With an additional 47,200 deaths prevented and 294 IS risks added, the benefit–risk ratio showed that 154 deaths would have been averted for every death that may have arisen from RoV [251]. This report was technically reviewed and adopted by the WHO Strategic Advisory Group of Experts (SAGE). While maintaining their status quo about the age restriction for the RoV (15–32 weeks), recommendations were made to support the evidence from Patel and co-workers’ reports, especially in regions where the benefit against mortality excessively outweighs the associated vaccine risks [252]. After that, several studies have shown a massive reduction in the incidence of RVGE hospitalisations in overaged children up to 18 years [253,254].

### 6.7. Underage

There are concerns about the neonates’ vaccination. According to the WHO recommendation, RoV should begin at no less than 2 months of age. This is the reason the 2Ds of RV1 is recommended to be administered at 2 and 4 months of age while the 3Ds of RV5 should be administered at 2, 4, and 6 months of age. However, few studies have highlighted the benefits of neonates’ vaccination with minimal associated relative risks.

A Phase IIa safety and immunogenicity trial of RV3-BB in New Zealand given at 3Ds (0–5 days, 8 and 14 weeks) showed 63% and 74% anti-rotavirus IgA seroconversion rate in the neonates and infants [172] (Table 1). This RV3-BB, produced from the parent nursery strain G3P2A [6], is generally referred to as the neonatal oral live RoV candidate. In its Phase IIb randomised, double-blind, placebo-controlled trial conducted in Indonesia, the vaccine efficacy against SRVGE and any SRVGE was 94% and 63% at 12 and 18 months in neonatal-schedule groups; in the same month, 77% and 45% efficacies were observed in the infant-schedule group [173].

Comparatively, the immunogenicity of RV3-BB appeared modest. When the safety and immunogenicity of RIX4414 live-attenuated HRV was analysed in Finnish infants, the vaccine induced a seroconversion rate from 50% to 88% after the first dose and 73–96% after the second dose, depending on the vaccine concentration [255]. The vaccine formulations are 10^4.7^ FFU/mL (with antacid), 10^4.1^ FFU/mL (with CaCO_3_ buffer), 10^4.7^ FFU/mL (with CaCO_3_ buffer), and 10^5.8^ FFU/mL (with CaCO_3_ buffer). Table 3 compares the vaccine response and stool shedding of RV3-BB and RIX4414 candidates. Apart from the improvement as compared with the RV3 parent strain, it is very convincing that RV3-BB has a considerable competitive vaccine “take” with the RIX4414 candidate. However, cumulative stool shedding of the vaccine strain is still a problem with this vaccine as this may pose the risk of transmission among children (Table 3). A comment from Vesikari in 2015 had shown that the inclusion of the vaccine shedding as evidence of vaccine “take” in Bines and co-worker, 2018, may not be appropriate and may prevent an independent comparison with other vaccines [256]. Furthermore, the insignificant effect of the first dose of RV3-BB as compared with the overall vaccine efficacy seemed not to purpose this vaccine exclusively for the neonates. Rather it can be given as well to infants and toddlers [256].

Previously, the M37 neonatal vaccine candidate tested in Finland had shown similar and comparable vaccine responses (47% and 76% IgA seroresponses to 10^4^ and 10^5^ PFU/mL, respectively) as observed in RV3-BB and RIX4414. However, the unconfirmed vaccine efficacy and high vaccine shedding (about 70% in infants receiving 10^5^ PFU/mL) discouraged further development of this vaccine candidate. Though the neonatal 116E strain genotype G9P[11] might have been developed into a licensed vaccine candidate, its preliminary efficacy in Indian infants was 55.1% against SRVGE, 57.2% against VSRVGE, and 55.6% against SRVGE with hospitalisation/rehydration therapy [60,183] (Appendix A). Nevertheless, this is an average vaccine performance compared with the RV1 and RV5 performances in the HICs.

The first BRV strain RIT4237 from serotype G6 failed to reduce the number of episodes of RVD in a single-dose with 5-day-old Finnish infants. However, it significantly decreased the clinical severity associated with RVD. This seemed not to correlate with the serological response but confirmed the presence of serum rotavirus antibodies, IgG, before the beginning of the seasonal epidemics [257]. Prior to this time, efficacy trials of RIT4237 in Finland when given at 1D to <1-year-old infants before the rotavirus epidemic season were well-tolerated with a protection rate of 88%, including protection against RVD caused by heterologous subgroup 2 strains [74]. Following a 2D regimen in the clinical efficacy of RIT4237 in Finland before rotavirus season, 82% vaccine protection was observed in <1-year-old infants. Not only this, but all forms of clinically significant diarrhoea were also reduced by 76% and seroconversion after vaccination was 53% with correlated clinical protection [258]. This vaccine candidate displayed heterologous protection of 72% against epidemic seasonal RVD caused by human serotype 1, 100% protection against human serotype 2 infection, and 100% protection against human serotype 3 infection [258]. Delem and Vesikari detected and quantified the serum antibody response to RIT4237 in <1-year-old infants and neonates vaccinated at 5 days old and the highest seroconversion of 79% was obtained in the infants, however, with ELISA-IgM and homotypic neutralisation assays, the antibody response rate was 31% and 45%, respectively, in newborns after immunisation [259]. The disadvantage of this heterologous vaccine candidate is that it can only be useful in a region where rotavirus season can be predicted because inconsistence efficacies in some other places thwarted the effort to produce this vaccine.

The withdrawn RRV-TV was also tested in neonates and infants that received 3Ds at 0, 2, 4 (A), 0, 4, 6 (B), or 2, 4, 6 (C). Neonatal regimens (A and B) failed to show signs of febrile reactions to the vaccine, unlike the infants that received the first dose at 2 months (C). Contrarily, by 5 and 7 months of age after the first dose, the IgA seroresponses to rotavirus were A (77–81%), B (70–86%), and C (100%), while neutralising antibody responses were A (74%), B (83–93%), and C (82–93%) [160]. Furthermore, group A showed a consistently lower neutralising antibody response to the RRV and HRV serotypes G1–G4 as compared with groups B and C. The highest neutralising antibody response was observed in group B [160]. This, therefore, shows that neonatal immunisation may reduce febrile reaction/reactogenicity, produce a competitive seroresponse, and probably dampen the IS formation, which is usually associated with aged-vaccination. Besides, successive vaccination, just at the neonatal–infant transition stage (group B), may just be sufficient to offer heterotypic protections against HRV serotypes. Though RRV-TV was withdrawn because of associated IS, which is also found in RV1 and RV5, there was no evidence to show that RV1 and RV5 could have done better than RRV-TV with neonatal immunisation.

### 6.8. Juvenile Immune System

The juvenile immune system, which consists of the innate and acquired immunities, is particularly associated with infant development [260] and plays a significant role in the mucosal immune system to enhance RoV efficacies. A report has shown a fundamental reduction in the efficacy of RoVs in the LICs due to some predicted intrinsic immunological and epidemiological factors [194]. Irrespectively, infants’ immune systems are generally immature and suffer RVI. Despite this, early natural RVI in infants before the first dose vaccine in LICs is one of the major underlining factors that modified the vaccine efficacy [194]. Early exposure incites the immune system to produce an anti-rotavirus antibody, and this, in addition to the maternal antibody and breast milk anti-rotavirus protein, renders the vaccine ineffective by preventing viral replication and antigen load in the gut [198]. However, the reverse is the case in the HMICs. This suggests a certain variation in the immunological parameters from different socio-economic settings. Notwithstanding, the infants’ immune system is fragile, immatured, and highly predisposed to various overwhelming infections. This fragile immune system becomes weakly responsive to most oral vaccines because of the compounding factors such as poor nutrition, co-infections, unhygienic environment, and low-quality microbiota, as frequently found in African, Asian, and some South American countries [194,261].

### 6.9. Tropical Enteropathy

Tropical enteropathy is an intestinal dysfunction caused by excessive colonisation of the Enterobacteriaceae. It is characterised by bacterial overgrowth at the proximal region of the small intestine accompanied by anaerobic fermentation of gut carbohydrates to produce hydrogen gas [262]. This poor intestinal condition can sometimes lead to environmental enteropathy (EE), characterised by several gastrointestinal dysfunctions such as intestinal inflammation, reduced intestinal absorption, and gut barrier dysfunction. It is otherwise called environmental enteric dysfunction (EED), which is reversibly determined by the exposed contaminated environmental conditions [263,264]. Apart from these, EED can be caused by an imbalanced supply of microelements like zinc and vitamins such as A and D [265]. This cluster can promote a poor immune response to oral and intranasal vaccine candidates, unlike parenteral vaccines [266]. The prognosis of this disease condition has been well studied with characterised physiological markers such as faecal calprotectin, neopterin, α-antitrypsin, and myeloperoxidase [267,268].

EED is more common in African children [269,270], and it is associated with a poor vaccine “take”. A significant report from Naylor and co-workers in Bangladesh showed that as high as 80% of the infants assessed usually manifested EE before 12 weeks of life and were characterised with malnutrition and systemic inflammation [266]. Furthermore, the infants showed 20.2% and 68.5% failure to OPV and RoVs, respectively, with extremely low antibody protection levels to tetanus (0%), *Haemophilus influenzae* type b (9%), diphtheria (7.9%), and measles vaccines (3.8%) [266]. Becker-Dreps and co-workers had already observed a negative correlation between the IgA seroconversion to the first dose of RV5 and various faecal biomarkers of infant EE [271]. This means that any intervention that ameliorates the infant EE condition in the LMICs may likely promote RoVE.

### 6.10. Histo-Blood Group Antigens

The peripheral genetic distribution of histo-blood group antigens (HBGAs) causes susceptibility variation to enteric infections caused by norovirus and rotavirus. HBGAs are among the host factors, which consist of complex pleomorphic glycosphingolipids or glycoproteins. These conjugated molecules form a plasma membrane receptor or membrane-tethered/secretion; while the former promotes viral infection, the latter antagonises. Structurally, the O-linked glycoproteins terminal oligosaccharides can either be -GalNAc (blood group A) or -Gal (blood group B). These terminal oligosaccharides are determined by the nature of the lactosamine chains (type 1 or type 2), which can either possess Lewis-type fucose/sialic acid or H-types (1–3) fucose (Figure 2). The terminal oligosaccharides and fucosylation are catalysed by glycosyltransferase and fucosyltransferase, respectively. Among these isozymes, the epithelial tissue and salivary gland α-1,2-fucosyltransferase 2 (encoded by *FUT2* allele) are called secretor transferases. Lewis antigen secretors, which arise from α-1,4-fucosyltransferase 3 (encoded by *FUT3* allele) catalysis are Le^a−b+^/Le^x−y+^, partial secretors are Le^a+b+^/Le^x+y+^, and non-secretors are Le^a+b−^/Le^x+y−^. Based on the ABO-locus encoding glycosyltransferase gene, H antigen can be further modified to produce the A antigen or B antigen. Type O individuals have an unmodified H antigen on their red blood cell plasma membrane, and this could be secretor (O)/non-secretor (O^−^) [272,273,274].

Gastrointestinal enterocytes express either sialylated or non-sialylated HBGAs, which serve as a decoy to RVI, and this phenotypic trait is directly quantifiable from the salivary glands. Animal rotavirus serotypes like P[1], P[2], P[3], and P[7] bind sialylated HBGAs for host cell attachment during infection. However, HRV genotypes such as G1, G2, G3, G4, G9, P[4], P[6], and P[8] and a majority of the animal rotavirus strains bind non-sialylated HBGAs. Available evidence showed that the VP8* of HRV strains selectively preferred the A-type HBGA membrane receptors [275]. Genotypes P[4] and P[8] generally attached to the same antigens of Le^b^ and H-type 1, while P[6] preferred the H-type 1 antigen only [276]. In the salivary binding assay, the P[8] VP8* of RV1 and RV5 bound secretors irrespective of ABO blood groups but failed to bind non-secretors, Le^b^, and H-type 1 antigens in the oligosaccharide binding assay [273]. The authors suggested the galectin-like fold structure of the vaccine VP8* had prevented the binding to the HBGAs. This is against the typical mucin-like fold structure needed for binding HBGAs.

A report has shown that strains like P[6] and P[11] have an age-specific host range with the ability of inter-species transmission. This feature is also found in serotypes like P[1], P[2], P[3], P[7], P[9], P[10], P[12], P[14], P[19], P[24], P[25], and P[28], which bind the A antigen in an oligosaccharide binding assay [274,277] (Table 4). These are possible because of the shared HBGAs ligands between humans and animals and perhaps a single evolutionary trend with a common ancestor with an animal host origin [274]. A broader analysis showed that the A antigen might be involved in the cross-species transmission of rotavirus because of its ability to bind sialidase-sensitive and non-sialidase-sensitive rotavirus strains [278] (Table 4). This is clear evidence of RVI as a zoonotic disease to humans.

Evidence from Nordgren and co-workers showed that P[8] rotavirus strains infect only Lewis and secretor positive children. In contrast, P[6] predominantly infects Lewis-negative children and to some extent, Lewis-positive children, irrespective of the secretor status [279]. Kambhampati and co-workers observed that secretors were 26.6 times more susceptible to serotype P[8] RVI than non-secretors [280]. This may be the reason there was a resistance against the G1P[8] genotype, including vaccine strains, because there is a high percentage of Lewis-negative children in Africa. This also explains why there is a prevalence of the P[6] rotavirus strain in Africa [279]. In Bangladesh, the vaccine of the P[8] genotype protected secretors and non-secretors equally, while unvaccinated non-secretors suffered infections from P[8] but were relatively protected against P[4] infections—the overall VE reduced from 56.2% in the secretors to 31.7% in the non-secretors [281].

Furthermore, following the 3Ds regimen of RV1 in Pakistan, Kazi and co-workers observed the lowest IgA seroconversion rate in the non-secretor infants. However, the highest rate was observed in the blood group O secretor infants [282]. This means that the HRV strains identify and bind HBGAs in a type-specific manner, which seems to be favoured by epitope B masking the H epitope to prevent the binding of P[8], unlike the A epitope that has no shield over the H epitope [276]. This attribute of the unmasking of the A epitope may further contribute to its roles in inter-species rotavirus transmission. This, therefore, means that host HBGAs are likely to affect individual susceptibility to genotype-based RVIs. The antigenic receptor distribution and phenotypic expression of HBGAs, therefore, have some significant impacts against the efficiency of RoVs “take” and invariably lower the effectiveness of these vaccines. These findings may be responsible for the lower vaccine efficacy observed in Africa and Asia.

### 6.11. Vaccine Interference from Co-Administration with Other Childhood Routine Vaccines

Co-administration with OPV has been shown to suppress RoV efficacy as indicated by the low level of serum anti-rotavirus IgA but not vice versa [283,284]. Contrarily, early studies about the safety and immunogenicity of RoVs have shown that Rotarix^®^ and RotaTeq^®^ were well-tolerated [285] and elicited a high immunogenic effect even when co-administered with other routine childhood vaccines [286,287]. It appears that OPV and RoVs are inseparable in the routine childhood vaccines because all RoV doses should be completed before 32 weeks of age while OPV is usually administered between 12 and 16 weeks of age. Precedently, an increasing number of RoV doses may compromise the overshadowing replicative effect of the OPV first dose to induce a mucosal immune response [283]. Ciarlet and co-workers reported an upward increase of ≥3-fold in serum anti-rotavirus IgA after the third dose of RV5 in Latin American infants to support the simultaneous tolerance of OPV and RoV [285].

In Bangladesh, Zaman and co-workers reported insignificant differences in the anti-rotavirus IgA seroconversion rate (56.5–66.7%) to RoV in a 2D Rotarix^®^, whether co-administered with OPV at 12 and 16 weeks or administered 15 days after OPV [288]. It is very surprising to observe further that seroprotection against polio was not affected either. Furthermore, it was reported that the third dose of RV1 has no interference with the measles-rubella vaccine (MRV) given at 9 months of age but rather improved the rate of anti-rotavirus IgA and IgG seropositivity and seroconversion to RVI without any case of IS or death [289]. Further work from Bangladesh has shown that the co-administration of 2Ds of Rotarix^®^ with mono-, bi-, and tri-valent OPV showed that OPV, irrespective of the valent, consistently reduced the anti-rotavirus antibody seroconversion rate in infants when concomitantly administered against non-concomitant administration [284].

In Mali, experimental results showed non-interference of RotaTeq^®^ with the measles vaccine (MV) and meningococcal A conjugate vaccine (MenAV) given simultaneously at 9–11 months of age but failed to conclude the non-interference with the yellow fever vaccine (YFV). Nevertheless, co-administration with RV5 significantly increased the anti-rotavirus IgA and IgG seroconversion and seroresponse rates as compared with the control (MV + MenAV + YFV only). At day 28 post-vaccination, 74.7% and 93.9% of the RV5 co-administration group had IgA and IgG ≥20 U/mL, while the control group had 58.9% and 76.1%, respectively [290]. These results significantly suggested the safety, tolerability, compatibility, and efficacy of RoV with other routine childhood vaccines.

Rotavac^®^ was compatible with childhood vaccines in Indian infants [291]. In a Phase III randomised, double-blind, placebo-controlled trial in Delhi, India, among 6-weeks-old infants, the following childhood vaccines OPV, DTaP, Hepatitis B, and *Haemophilus influenza* type b were co-administered at 6, 10, and 14 weeks of age with Rotavac^®^. Without any non-inferiority immune response, the seroprotective level against all these childhood vaccines was between 92% and 100% [291]. The geometric mean concentration (GMC) for pertussis was 18.5 vs. 17.7 between Rotavac^®^ + other vaccines and placebo + other vaccines, respectively, providing a GMC ratio of 1.0 (^Placebo^/_Rotavac_^®^). Non-interference of Rotavac^®^ was supported because the GMC was a >0.5 threshold [291]. This suggests that the vaccine is moderately compatible and the efficacy is the same as found in most of the LICs. The vaccine was well-tolerated without any associated adverse events. The five deaths reported during the follow-up in this study were not related to the vaccine. Again, there was no reported case of IS until after 1 year of age.

RotaSIIL^®^ as well has shown non-interference with childhood routine vaccines in 6–8-weeks-old Indian infants. In Phase III multicentre open-label randomised controlled study, 3Ds of RotaSIIL^®^, 2Ds of Rotarix^®^, and a dose of placebo were concomitantly administered with other childhood vaccines at 6, 10, and 14 weeks of age. Seroprotective/seropositivity rates among the vaccinees ranged from 97% to 100% [292]. Likewise, the GMC ratio (^BRV-PV^/_Rotarix_^®^) after four weeks of final vaccination ranged from 0.93 to 1.04, which supported the non-interference vaccine hypothesis [292]. Similarly, a Phase III clinical study was reported for the safety, immunogenicity, and lot-to-lot consistency of 3Ds of RotaSIIL^®^ with 2Ds of Rotarix^®^ and a dose of placebo in 6–8-weeks-old Indian infants, who had earlier received HepB and OPV vaccines at birth. The IgA GMC ratios among the lots A, B, and C considered ranged from 0.99 to 1.07 [293]. In terms of IgA seropositivity to RoVs, there was a 46.98% rate with RotaSIIL^®^ vs. 31.12% with Rotarix^®^ [293]. All forms of reactogenic effects remained the same across the lots and RotaSIIL^®^ vs. Rotarix^®^ combined vaccines. No cases of IS were detected. These results have shown the competitive safety and immunogenic profiles between RotaSIIL^®^ and Rotarix^®^.

Many other studies have shown that LAORoVs could not interfere with other childhood routine vaccines, which are either administered parenterally or orally. The co-administration of RIX4414 and OPV/inactivated polio vaccine (IPV) in 5–10-weeks-old South African infants using two vaccination schedules, 6–10 and 10–14 weeks, showed no effect on the seroprotection offered by the OPV/IPV against poliovirus serotypes 1, 2, and 3 (98–100%) [294]. In this Phase II double-blind, placebo-controlled trial, the observed anti-rotavirus IgA antibody seroconversion rate at 6–10 weeks in the vaccination schedule was 36–43% and for 10–14 weeks in the vaccination schedule, it was 55–61% [294]. The seroconversion rate appeared to improve with the dose but was still within the average vaccine performances in Africa.

In Europe, Vesikari and co-workers examined the immunogenicity and safety of Rotarix^®^ co-administered with routine infant vaccines (Infanrix hexa™, Infanrix quinta™, Meningitec™, and Prevnar™). Rotarix^®^ maintained a remarkable seroconversion rate among the infants (6–14-weeks-old) that concomitantly received these vaccines with a 94.6% and 92.3% rate in Finland and Italy, respectively, one month after the second dose. These two countries followed three and five months of vaccination schedules. With two months post-vaccination, the seroconversion rate in Germany with two and three months vaccination schedules was 82.1%, France with two and three months vaccination schedules was 84.3%, the Czech Republic with three and four months vaccination schedules was 84.6%, and Spain with two and four vaccination schedules was 85.5% [295]. After the third dose of each of the childhood routine vaccines, there was no seroprotective/seropositive rate to each of these vaccines, which was <90% in each of the countries [295]. The overall anti-rotavirus IgA antibody seroconversion rate was 86.5% against 6.7% in the placebo group between one and two months after the second dose without any associated reactogenic events [295].

Among the USA infants range from 6- to 12-weeks-old, 2Ds of RIX4414 either co-administered with Pediarix, Prevnar, and ActHIB within two, four, and six months vaccination schedules or administered separately at three and five months vaccination schedules showed no impact on these childhood routine vaccines [296]. Each of the two administration groups achieved ≥ 97% of the infants showing antibody responses to all other childhood vaccines [296]. Still in the USA, Rodriguez and co-workers examined the concomitant administration of childhood routine vaccines with RotaTeq^®^ or placebo only between 2002 and 2003 among 6–12-weeks infants. A similar seroprotection rate was observed against all the routine vaccine antigens in the RotaTeq^®^ and placebo groups except for a moderate decrease in the antibody against the pertactin component of the pertussis vaccine in the vaccine group [297]. The authors further stated that the 35% antibody titre in the vaccine against 60% in the placebo to pertactin did not meet the noninferiority criterion. However, the efficacy of RotaTeq^®^ against RVGE of any severity remained high, at 89.5%. By comparing this efficacy with the REST evaluation for RotaTeq^®^ (74% against G1–G4 RVGE of any severity and 98% against G1–G4 SRVGE), concomitant administration with other vaccines has no impact on the efficacy of RotaTeq^®^. However, a case of IS in the vaccine group was reported. This occurred in a 5-month-old female 20 days after the second dose of RotaTeq^®^ and the completion of other vaccines. Apart from this, the vaccine was well-tolerated without any significant adverse events when administered on the same day with other childhood routine vaccines [297].

Further, though RotaShield^®^ was no longer in use, its assessment with OPV co-administration under different buffer conditions in the USA infants, who were 6–16-weeks-old, showed that concomitant administration with OPV had no impact on the vaccine efficacy, rather the buffer systems [298]. Moreover, the RRV-TV did not interfere with the neutralising response to OPV serotypes 1, 2, and 3. The effect of co-administration with OPV was further investigated with RRV-Serotype 1 and RRV-TV among 5–25-weeks-old USA infants. There was no interference observed by these vaccines. There was no significant difference in the seroprevalence and GMT of the antibody against OPV serotypes 1, 2, and 3. However, as the OPV doses increased, anti-rotavirus IgA GMT with RRV-S1 also increased, but the opposite was the case for the RRV-TV. The neutralising antibody and seroconversion rate to RRV, G2 (DS-1), G3 (P), and G4 (ST-3) appeared significantly higher with RRV-TV than RRV-S1. Nevertheless, the neutralising antibody and seroconversion rate to G1 (Wa) was higher with RRV-S1. Though the sample size was small, the authors still concluded that concurrent OPV doses appeared not to have a significant effect on RVGE rates among rotavirus vaccinees [299]. These results showed homotypic and heterotypic protections by RRV-S1 and RRV-TV, respectively, and these seemed not to be interfered by the concurrent administration with OPV. However, a slight strain interference, by suggestion, could have warranted a moderate reduction in the IgA post-vaccination GMT titres observed with the RRV-TV.

In Thai infants, however, RRV-TV seemed to prefer IPV to OPV. After one month of the first dose of RRV-TV at 2 months of age, 37% of the infants that received RRV-TV + IPV had seroconverted compared with 10% from RRV-TV + OPV [300]. Further, 74% of the infants with IPV showed a neutralising antibody seroconversion rate to RRV-3 after the first dose of RRV-TV compared with 39% with OPV. The second and third dose of RRV-TV seemed not to improve these conditions. At the seventh month post-vaccination, the neutralisation antibody to human serotypes 1, 2, and 4 appeared moderately higher with IPV than the OPV group. Vaccine interference was not observed. The seroprotective antibody to polio serotypes 2 and 3 was similar in the RRV-TV + OPV and OPV alone; however, a slight antibody reduction to serotype 1 was observed at the fifth and seventh months of age [300]. This observation suggested that OPV may be interfering with RRV-TV; however, increasing the vaccine dose/titre may compensate for this reduction.

Further work on the immunogenicity and reactogenicity of RRV (strain MMU 18006) co-administered with OPV and DTaP in the USA infants, who were 8–12-weeks-old, showed 56% of infants with a 4-fold increase in IgA seroconversion and 62% with a 4-fold increase in the neutralising antibody to RRV [301]. There was no interference imposed on the OPV either. No less than 60% of the infants showed a 4-fold increase in the neutralisation titre to at least one of the three polio serotypes three–five weeks post-vaccination [301]. Therefore, RRV was not associated with any adverse events and co-habited the infants’ gut successfully with the OPV and DTaP vaccines. In Pakistan, RRV was co-administered with OPV or DTaP or IPV in 6-weeks-old infants. At three weeks post-vaccination, 50% of the RRV + OPV infants had a 2–4-fold increase in the neutralising antibody titre against rotavirus, while 22% from RRV + DTaP and 20% from RRV + IPV showed an increased neutralising antibody [302]. Interference from RRV was unlikely as 81%, 67%, and 59% successfully seroconverted to IgG antibody against polio serotype 1 [302]. The low rate antibody neutralisation against rotavirus observed showed that an increased vaccine dose/titre might improve the antibody neutralisation. Notwithstanding, the performance of RoVs in the LICs is usually on an average.

With RIT4237 among the 12-weeks-old infants, co-administration with OPV led to a reduced seroconversion rate to rotavirus among the vaccines, while the antibody response to polio serotypes 1 and 3 remained unaffected [303]. Immunisation with RIT4237 alone showed a 73% seroconversion rate among the vaccinees, and the seroprotection against poliovirus serotypes 1 and 3 remained yet unchanged [303]. Similarly, Giammanco and co-workers reported a dramatic reduction in the seroconversion rate among the 12-weeks-old infants induced with RIT4237 + OPV. The attempt to overcome this by increasing the vaccine dose proved abortive; however, there was no observed interference on the immune response to the polio vaccine [304].

### 6.12. Sequential Vaccine Doses

Factors such as the differential vaccine doses received by the infants may probably not have a significant effect on the ultimate protection by the vaccine. Ideally, all except Rotarix^®^ are given in 2Ds, and other licensed LAORoVs are given in 3Ds. The overall reduction in risk associated with incomplete immunisation with one or two doses of the RV5 vaccine appeared to be similar in infants with complete 3Ds [239]. However, a significant difference may be observed in HICs when complete 3Ds of RV5 is compared with 1D [305,306]; however, in Africa, the effectiveness of the complete doses of RV5 was generally low, let alone the incomplete doses [249]. Similarly, significant differences are likely to be observed in selected countries when the VE of 2Ds of RV1 is compared with the 1D incomplete regime. Such countries include Spain [307], Brazil [186], and many African countries [308,309] (Appendix A). Surprisingly, incomplete doses have produced more effectiveness than complete doses, as observed in Africa [308,309]. An increased number of doses may be considered in vaccine implementation in impoverished countries of Africa and Asia, where the burden of RVI is very high, but this may not play any significant role in elevating the vaccine efficacy in the developed countries.

In the analysis of RV1 efficacy over two years against RVGE in Malawi, the efficacy of 2Ds was compared with 3Ds. At the end of the first and second year of follow-up, only marginal differences were observed without any significant differences in terms of the concentration of the serum anti-rotavirus IgA and rotavirus episode prevented/100 infants vaccinated/year [179]. Similarly, Madhi and co-workers observed no significant difference between 2Ds vs. 3Ds of RV1 in South Africa and Malawi [130]. In both countries, a greater proportion of diarrhoea-related cases were prevented with higher efficacy using 2Ds rather than 3Ds regimens [130]. However, in 2012, Madhi and co-workers carried out a randomised, double-blind, placebo-controlled trial to compare the efficacy and immunogenicity of 2Ds and 3Ds of RV1 in South Africa and discovered a higher point efficacy over two seasons of RVI in the 3Ds than the 2Ds [192]. Apart from the fact that efficacy increases with severity, there was a 44% reduction in all-cause SRVGE in 3Ds but no significant reduction with 2Ds of RV1 [192].

In a similar assessment in Ghana, the third dose of RV1 resulted in an increased seroconversion rate and IgA titre geometric value [310]. This appears promising and may be of huge benefit in the LICs, but due to the small sample size and lack of follow-up protection analysis, it may be difficult to translate this to a public health benefit. Notwithstanding, larger sample sizes have been assessed for vaccine efficiency in South Africa and Malawi with 2Ds and 3Ds pooled samples and discovered a more significant efficacy of Rotarix^®^ against rotavirus infection, especially after the first year of follow-up over the second year of follow-up [179,192].

In India, Kompithra and co-workers in 2014 reported lower seropositive responses in babies receiving 3Ds or 5Ds of RV1 along with scheduled childhood vaccines [311]. This low immunogenicity against the G1P[8] vaccine was associated with the natural wild-type infection caused by G10P[11] in hospital-born children, where the high rate (>50%) of anti-rotavirus IgA baseline seropositivity was reported at 6–12 weeks of age [226,312]. Compared with the placebo with a 6.3% seroconversion rate, Narang and co-workers observed a 58.3% seroconversion rate to 2Ds of RV1 two months after the last dose [226]. With Lokeshwar and co-workers, 83% of the infants exhibited at least a 3-fold increase in the seroconversion level of the serum anti-rotavirus IgA antibody to 3Ds of RV5 [313]. Together, the vaccines were immunogenic, well-tolerated with a good safety data profile. In Pakistan, however, the immunogenicity of RV1 failed to increase significantly with 3Ds at 6, 10, and 14 weeks (25.8 U/mL IgA at a 36.7% seroconversion rate) compared with 2Ds at 6 and 10 weeks (24.0 U/mL IgA at a 36.1% seroconversion rate) or 10 and 14 weeks (24.4 U/mL IgA at a 38.5% seroconversion rate) [314].

## 7. Problems Associated with the Live-Attenuated Rotavirus Vaccines

The live-attenuated oral rotavirus vaccine essentially contains active vaccine particles, which means it can replicate in a favourable host cell. While producing this vaccine candidate, several cellular passages were done to minimise the replicative efficiency of the viral particles. In classical virology, when viral particles are passaged multiple times, it becomes weaker and less immunogenic. These activities bring the viral particles to be attenuated while still active. RoVs are popularly passaged in cells like AGMKC and MA104 for vaccine production and are administered orally. Infants’ gut provides a suitable environment for the activation of viral replication following successful trypsin activation of the spike protein VP4 found on the virus outer layer. The virus becomes active and infects the epithelial enterocytes for attachment and subsequent transcytosis. This brings about changes in body physiology, and several reports have suggested a few clinical effects of the LAORoV.

### 7.1. Live-Attenuated Rotavirus Vaccines Cause Intussusception

Intussusception is a rare clinical side effect of RoVs. It is the intestinal convolution against the free flow in the gut caused by the invagination of the distal into the proximal ileum along the intestinal tract. Clinical presentations of IS include the three basic triad symptoms: vomiting, abdominal pain, and rectal bleeding/bloody stool [315,316]. Reports by WHO in 2002 on the global perspective of the incidence, clinical presentation, and management of acute IS in infants and children presented Vietnam with 472–722 natural cases of IS annually in children <12 months of age, which are presented to hospitals between the period 1995–1999, followed by China with 492 cases [317]. By 2013, IS occurs at a mean incidence of about 74 cases per 100,000 children < 12 months and differs geographically. Asia has the highest occurrence of IS with Vietnam and South Korea, having as high as 302 and 328 per 100,000 infants, respectively. However, the highest fatality rate is in Africa (9%) compared with the other continents (<1%) [318,319]. The data reported here were a pool from 82 studies across the globe with a total case of 44,454 [319]. The epidemiological occurrence of intussusception is very low. Jiang and co-workers compiled childhood IS and observed a rate of 13–37 per 100,000 in infants <2 months to toddlers >12 months of age, but the peak incidence usually occurs between 4 and 7 months of age with a rate of 97–126 per 100,000 [319]. The peak incidence of IS between 4 and 7 months of age is an indication that RoV should be administered early enough to minimise the risk of IS due to vaccine since most of the reported vaccine-associated IS events occurred between one and seven days after the first dose or 21 days after the second dose of RoV [251,320,321] (Appendix A).

There was no clear association of IS with the seasonal distribution of RVD [322]. However, the causes of IS may be more complex than what is known, and its seasonal occurrence is regionally unique. It may be related to the peak incidence of hospitalised AGE/diarrhoea and respiratory tract infection (RTI) [323,324] or not [325]. Certain adenoviral infections, hypergastrinaemia, and few enteric viruses/pathogens/parasites may be more responsible rather than feeding habits, genetic predisposition, and acclaimed vaccines [318,326]. In other reports, a wide range of risk factors has been implicated. Intussusception is more likely common in boys than girls, children with nonsolid foods diet, babies with formula-feeding, babies with mesenteric adenitis, black race ethnic babies, and babies suffering from various viral infections and diarrhoea [317,327].

RRV-TV (RotaShield^®^) was withdrawn from the market by October 1999 because of an increased attributable risk of IS (1 in 4670–9474 vaccinated children), which occurred 3–14 days after the administration of the first dose of the vaccine [59,63]. Between 1998 and 1999, 15 cases of IS were reported as a vaccine adverse event among the USA infants who received RRV-TV [328] but follow-up studies of this birth cohort failed to reveal any evidence of increased IS rates in infants between the 1998 and 1999 period [329,330]. Hence, the aetiology of IS was examined and it was concluded that the vaccine might have triggered the IS occurrence in some infants as there was no evidence of a direct link between IS and RVI [317]. Irrefutably, all the current live-attenuated vaccines have been reported to cause a small increase in IS [331], with Rotarix^®^ and RotaTeq^®^ reportedly having an occurrence of 1–2 cases per 100,000 vaccinations. The risk attributable to these vaccines is in excess of 1–7 cases per 100,000 vaccinations [332] as compared with the RRV-TV vaccine with a risk ratio of 22.7 cases after the first dose, however, the reports differed from country to country [59,133,333] (Appendix A).

Serotonin is a neurotransmitter that may influence intestinal neuromuscular coordination and increase the risk of IS [317,334]. The presence of the rotavirus neurotoxin, NSP4, in the RoV may be of concern in the intestinal release of serotonin, and this interaction deserves more research insight. The evidence failed to show an overall increase in the relative risk of IS associated with the current RoVs, however, there is no statistical evidence to rule out the likely occurrence of IS following the first dose of either Rotarix^®^ or RotaTeq^®^ [320,321,335] (Appendix A). In Mexico, RoV caused 1 case of IS per 51,000 vaccinated infants (about 41 excess hospitalised IS plus 2 deaths), while in Brazil, the statistics were 1 per 68,000 vaccinated infants (about 55 excess hospitalised IS plus 3 deaths) and a combined annual excess of 96 cases of IS and 5 deaths [133]. Again, with RV5 in the USA, an excess of 33 cases per year were inferred [331]. From the benefit-risk analysis, the use of RV1 would have averted 663 deaths and 11,551 hospitalisations due to RVI in Mexico and 640 deaths plus 69,572 hospitalisations in Brazil among children <5 years [133].

Further, RV1 causes about 21 IS hospitalisations in England but prevents 25,000 cases of gastrointestinal infections [321]. Yen and co-workers compared the occurrence of IS before and after the introduction of RoV in the USA. It was discovered that the rate of IS among children between 8- and 11-weeks-old increased from 2007 to 2009 (with 11.4 per 100,000, 12.2 per 100,000, and 11 per 100,000, respectively) after the vaccine introduction as compared with 2000–2005 (6.9 per 100,000) before the vaccine introduction [336]. Nonetheless, the benefits accrued to the rotavirus vaccination is of magnitude impact to public health. Despite all these, the WHO’s Global Advisory Committee on Vaccine Safety still maintains their status quo that RoVs ought to commence between 6 and 15 weeks. However, the first dose after 15 weeks may still be carried out, especially in some LICs and MICs where vaccine availability appears to be a problem. After all, the abounding benefits in these vaccines against RVI supersede the traceable occurrence of vaccine-associated IS events.

### 7.2. Incidence of Adventitious Contaminants in the Live-Attenuated Oral Rotavirus Vaccines

Porcine circoviruses (PCVs) are non-enveloped viruses with single-stranded circular DNA that have not been reported to infect humans or cause a cytopathic effect in cell cultures [337,338]. However, PCVs have been found in the vaccine formulation of the two globally licensed rotavirus vaccines [339,340]. Rotarix^®^ was found to contain a full-length PCV-1 genome that is particle-associated and infects PCV-free porcine kidney cell (PKC-15), but RotaTeq ^®^ was found to contain barely detected small genome fragments of PCV-1 and PCV-2 [340]. Sadly, Spain withdrew the Rotarix^®^ vaccine for concern of PCV-1 contamination, and France has constantly lamented over the vaccine safety since 2015 because of the associated IS [341]. Four years after the license of Rotarix^®^, regulators in the USA advised doctors to temporarily suspend the use of this vaccine because of the suspected impurities, which GSK indeed admitted. PVC-1 [342,343] and PCV-2 [344] are generally a global infection of swine; however, while PCV-1 may not cause infection in animals, PCV-2 maybe a helper virus for co-infection [337]. Sometimes, a cellular fragment including nucleic acids from the Vero cells used in passaging can be found in the vaccine products; however, this appears as a limiting factor. Whether this may cause a significant effect is not known for now, but there is PCV genomic material transferred each time an infant is vaccinated. The cumulative effect seems to be extraneous but may induce an unusual immune response. Despite this possible effect, the recommendation to use these vaccines remains, and with stringent quality control in the manufacturing process, a PCV-free vaccine can be made possible.

### 7.3. Live-Attenuated Oral Rotavirus Vaccines Reassort

Live-attenuated rotavirus vaccines reassort to virulence strains [345] in two ways. Firstly, with the circulating field strain and secondly, within attenuated rotavirus components of the polyvalent vaccine such as RotaTeq^®^ and RotaSIIL^®^. In either of the cases, a new strain emerges with a different partial or complete genomic constellation, which can be more virulent/less virulent. A few instances have been reported in the post-vaccination period, for example, the discovery of vaccine-derived G1P[8] from RotaTeq^®^ [346,347] and Rotarix^®^ [348,349] and heterotypic G2P[4] from RotaTeq^®^ [350]. This poses a serious threat against vaccine efficacy [351]. This is, in fact, one of the main risks associated with the use of replicating live-attenuated oral/intranasal vaccines and underlines the need for a thorough evaluation and impact of this against the public health benefit. The implication of this is the emergence of new and diverse strains, which can be resistant to the vaccine. The use of alternative, non-replicating vaccine candidates, is continually gaining more insight into the clinical trials. Eventually, perhaps, this may complement the LAORoV and reduce the live-attenuated vaccine pressure and chances of reassortants.

### 7.4. Live-Attenuated Oral Rotavirus Vaccines Require Cold Storage Facility and Transportation

All the LAORoVs are cold-chain storage vaccines, even the cost-effective thermostable RotaSIIL^®^ best functions in cold storage compared with an ambient storage condition [19,166]. This procedure involves various logistical issues such as storage, transportation, and distribution, which may be difficult to implement in LICs [352,353]. Inadequate storage and transportation facilities for these cold-chain vaccines could lead to massive wastage of the vaccine products [55,166]. This condition of storage and transportation singlehandedly determines the potential costs of the introduction and implementation of RoVs. Therefore, policymakers need to not only consider the vaccine cost but also the cost of a cold storage facility for transportation, especially into the rural settings where there is no electricity. In addition to these, the cost of cold chain expansion and distribution, staff empowerment, and delivery services cannot be ruled out in the implementation of these vaccines.

### 7.5. Formulation, Processing, and Packaging of the Live-Attenuated Oral Rotavirus Vaccines

The mode of the formulation, processing, and packaging of RoVs affects the vaccine choice, acceptability, ease of vaccine completion, and cost-effectiveness. These factors may indirectly affect the efficacy and effectiveness of RoVs as well. As the vaccines are live-attenuated, maintaining the antigenic biological structures is very paramount to vaccine immunogens. Sucrose and glycine in sodium bicarbonate buffer are among the pre-packed vaccine constituents. These respectively raised the density cushion for the antigenic protein to form intermolecular H-bonds that maintain the antigenic position during lyophilisation. Thus, the antigenic quaternary structure of the vaccine is maintained in the excipients. Citrate bicarbonate buffer, as a diluent, is used for reconstitution prior to immunisation to prepare an equilibrium pH condition. Though lyophilisation maintains the antigenic shape, prevents unwanted chemical reactions among the vaccine components, and prolongs the shelf-life depending on the storage temperature [354], it can agitatedly disrupt the stability of the vaccine’s bio-components during freeze-drying and lead to about a 40% loss of viral viability [355].

Recently, RotaSIIL^®^ was re-presented as a liquid ready-to-use vaccine and launched in India. The immunogenicity and lot-to-lot consistency were compared with powdered/lyophilised reconstituted RotaSIIL^®^ in 6–8-weeks-old Indian infants. This Phase II/III open-label randomised study involved three lots of liquid RotaSIIL^®^ (LR) and one batch of powdered RotaSIIL^®^ (PR) reconstituted in 2.5 mL buffer diluent—25.6 g of sodium bicarbonate and 9.6 g of citric acid per litre. Each vaccine, kept between 2 and 8 °C, contained ≥10^5.6^ FFU each of G1, G2, G3, G4, and G9 serotypes per dose of 2.0 mL administered at 6, 10, and 14 weeks of age. The GMC of anti-rotavirus IgA was 36.35% for LR vs. 30.51% for PR (GMC ratio = 1.19), the seropositivity ≥20 U/mL was 60.41% for LR vs. 52.75% for PR (GMC ratio = 7.66), and the GMC ratio of anti-rotavirus IgA among the three lots was 1.34 for lot A vs. B, 1.22 for lot A vs. C, and 0.91 for lot B vs. C [356]. Non-inferiority of the LR to PR was proven because the GMC ratio was >0.5, and the lot-to-lot consistency was also established because the GMC ratios were within 0.5–2.0 [356]. The vaccine lots showed no interference with other childhood vaccines and were well-tolerated without any related associated adverse events. This observation showed that the safety and immunogenicity of RotaSIIL^®^ were not significantly changed despite a change in the formulation and packaging. A nominal increase observed in the immunogenicity of LR against PR may probably be due to the liquid-formulated version of the vaccine. This is probably another landmark in the RoV production. This will improve the available vaccine option among the country demands, however, this development does not seem to improve on the problem associated with cold-chain storage and transportation.

The efficacy of RotaShield^®^ had been examined with different buffer systems—small and large buffer systems. The small buffer system contained 64 mg of sodium bicarbonate plus 24 mg sodium citrate in 2.5 mL of water, while the large buffer system contained 400 mg of sodium bicarbonate in 25 mL of soybean formula [298]. The IgA seroconversion rate for infants receiving the small-volume buffer with RRV-TV + OPV was comparable to infants receiving the large-volume buffer (45% and 49%, respectively). These seroconversion rates were significantly higher than those for infants that received RRV-TV with no buffer (23%) or OPV alone (13%) [298]. This means that the buffer system plays some significant roles with regards to vaccine efficacy. Not only this, co-administration of OPV had no impact on the vaccine efficacy. It is therefore very important to train the health workers and vaccinators regarding the storage, preparation, and administration of RoVs for effective vaccine coverage and elimination of biological bias in data analyses.

Apart from the safety and efficiency of RoVs, policymakers do examine the mode of formulation, processing, packaging, presentation, and ease of dosing during the cost of introducing and implementing the vaccines. These accessory costs, together with the vaccine cost, will be of economic consideration to self-financing or countries transiting from GAVI financial support [55]. In a technical model put forward by Pecenka and co-workers to re-evaluate the cost and cost-effectiveness of RoVs in three GAVI-supported LICs—Bangladesh, Ghana, and Malawi—these accessory costs were considered. It was concluded that the incremental health system cost per dose depended on various factors such as vaccine presentation, vaccine volume, cold chain storage/transport requirements, delivery process, and health staff training [55]. From the numerical integration of the vaccine dose, vaccine coverage, vaccine wastage, full course vaccine efficacy, and waning, the cost and cost-effectiveness of Rotarix^®^, Rotavac^®^, and RotaSIIL^®^ were analysed. Rotarix^®^ remained the least costly but the most cost-effective vaccine in these countries because of its lowest incremental health system cost per dose. The Rotarix^®^ 2Ds regimen, low vaccine wastage, vaccine cost, and vaccine package are the important criteria that contributed to its cost-effectiveness.

### 7.6. Rate of Viral Passaging May Induce Vaccine Adverse Events/Reactogenicity

Another minor but important factor is the frequency of passaging, which can affect the attenuated phenotypic features. The more the passaging, the more the attenuation from the precursor/parent strains; for example, the precursor strain for Rotarix^®^ (RIX4414) is the 89-12 rotavirus strain which was passaged 43–45 times before the final lyophilised vaccine vial. Perhaps the mode of vaccine preparation might play a role in adverse events/reactogenicity. Compared with the placebo, a significant increase in fever on day three–five after immunisation with the RRV-TV and 89–12 vaccine candidates were reported with ≥38.1 °C [357,358], however, this tendency was uncommon with RIX4414 (Rotarix^®^).

## 8. Impact of Rotavirus Vaccines on the Genotype Distribution Pattern

The massive rollout of RoVs in recent years, especially Rotarix^®^ and RotaTeq^®^, has resulted in strain selection due to vaccine-induced selective pressure [359]. In Australia, for example, before vaccine introduction, G1 was the most prevalent genotype followed by G2, G4, G3, and G9 [359]. After vaccine introduction, G1, G2, G3, G4, G9, and G12 were reported [359]. The emergence of G12P[8], equine-like G3P[8], and the increased frequencies of G8 and G10 represented the diverse population of genotypes circulating following vaccination in Australia [359]. The emergence of G12P[8] and equine-like G3P[8] was identified in regions with more than 90% coverage for the RotaTeq^®^ vaccine [163,360]. Unfortunately, lower vaccine efficacies have been reported against these emergent strains [359,361]. The emergence of G2P[4] strains, which are heterotypic DS-1 genogroup emerging human strains, has been reported by regions with the increased use of Rotarix^®^ (a Wa-like genogroup) [359,362]. Due to its heterotypic DS-1 genogroup human strain, consistent low vaccine efficacies have been reported against G2P[4], especially in the LMICs where strain diversity is very common [193,363].

Patel and co-workers associated a relatively lower performance of RV5 in Nicaragua to the prevalence of the G2P[4] genotype, which constituted an 88% occurrence in the samples used for the vaccine assessment [239]. However, Correia and co-workers reported about a 77% VE of RV5 against this strain [242], but in the USA and Finland, 88% was reported [165], though with a lower sample number as compared with the Nicaragua study.

The tendency of RV1 efficacy is very high against strains with P[8] antigens such as G3P[8], G4P[8], and G9P[8] [127]. Besides, this vaccine has displayed a high level of protection against the emerging G9P[4] genotype in Mexico with about a 94% VE and a conclusion that the emergence of this genotype in Mexico nationwide may not have been associated with the vaccine selectivity/pressure [364]. Evidence from the high mortality rate countries showed homotypic and heterotypic cross-protections by RV1 against severe rotavirus infections over two years of monitoring immunisation. About 59–93% VEs against the circulating serotypes including G1P[8], G9P[8], G3P[8], G2P[4], and G9P[6] in Bolivia [175,240] and Botswana [204] have been reported, but in Malawi, 51–82% VEs were reported against any P serotype of G1 and G12 or any G serotype of P[6] and P[8]; however, a significantly lower VE was reported against G2P[4] or any G serotype of P[4] [140]. In South Africa, about a 71% VE was recorded against the dominant strain G12P[8], 62% against any homotypic or partially heterotypic strains (either G or P protein similar to the vaccine strain), and 52% against any fully heterotypic strains (totally different from the vaccine strain) [365]. Generally, the effectiveness of the vaccines against the circulating strains, either homotypic or heterotypic strains, is high in Europe [183,366], North America [129,184], and Central America [185,364], moderately high in South America [175,186], and moderately low in Asia [57,131,181] and Africa [137,187] (Appendix A).

Many regions in Africa are generally known for diverse strains of rotavirus, and this may be one of the reasons for the low VE. Generally, the pre-vaccination genotype strains in most parts of Africa are G1P[8], G2P[4], G2P[6], G9[8], G12P[8], G8P[4], G1P[6], and G3P[6] with minor atypical strains. Further, the following genotypes have been observed after the introduction of the vaccine: G1P[8] being the most frequent, followed by G2P[4], G8P[4], G12P[8], G2P[6], G4P[8], G3P[6], and G9P[8], coupled with a few low incidences, mixed serotypes, and atypical genotypes such as G1P[4], G2P[8], G9P[4], and G9P[12]. These are resource information from fifteen countries across the eastern and southern Africa regions [367,368].

A systemic review and meta-analysis of the distribution of the rotavirus strains and strain-specific effectiveness of the vaccines showed that the effectiveness of Rotarix^®^ against homotypic, partly heterotypic, and fully heterotypic strains in the HICs was 94%, 71%, and 87%, respectively, while in the MICs, 59%, 72%, and 47% were reported, respectively [369]. For RotaTeq^®^, the effectiveness against the homotypic, partly heterotypic, single-antigen vaccine and non-vaccine strains was 83%, 82%, 82%, and 75%, respectively, while in the MICs, a 37% VE was observed for the partly heterotypic strains, 70% for the single-antigen vaccine strains, and 87% against the single-antigen non-vaccine strains [370]. Without any trace of persistent specific strain or vaccine-induced selective pressure, this observation showed that the effectiveness of both vaccines was the same with homotypic and heterotypic rotavirus strains [370]. Evidence continues to show that G2P[4] is common among the countries using RV1, while G1P[8], G2P[4], and G3P[8] are common in the countries using RV5 [371,372]. While some reports could not conclude the association of rotavirus fluctuations with vaccine pressure [373,374], others disjointed the relationship and assigned evolution, including novel genotype constellations, to natural selection through reassortments [361,375]. Nevertheless, vaccine introduction has caused a considerable drift in the rotavirus natural genotype fluctuations [363,372]. This incidence requires adequate and close monitoring to ensure the enduring effectiveness and efficacy of RoVs.

## 9. Vaccine- and Natural-Acquired Protections against Rotavirus Infections

Apprehension looms about the persistent nature of vaccine-acquired protection against RVI beyond the first year of follow-up, especially in the LMICs. From Table 2, it is very conspicuous that the VE decreases in the second year of follow-up in Malawi [180] and Nicaragua [179], unlike the developed countries where a higher VE has been observed in the two years of follow-up as well as against specific circulating strains (G1–G4 and G9) [135]. For example, in Finland, RotaTeq^®^ reduced RVGE hospitalisations and EV, irrespective of the circulating serotype, by 93.8% for up to 3.1 years after the last vaccine dose. Further, a similar trend was observed within the following age groups: 4–11 months = 93.9% reduction, 12–23 months = 94.4% reduction, and 24–35 months = 85.9% reduction, even AGE caused by rotavirus or other agents was reduced by 62.4% [183]. This is indeed a high vaccine efficacy with sustainable protection against rotavirus-related hospitalisations and EV without any occurrence of IS.

Repeated natural infection, whether symptomatic or asymptomatic, with rotavirus wildtype strains, has been shown to induce some level of protection against subsequent RVIs. The overall protection offered by the natural infection against symptomatic reoccurrence infection has been estimated as 93% with a tendency of heterotypic protection [98,376]. It has been shown that primary natural infection usually induced the highest neutralising antibody to human serotype 1 and then serotype 3, but a low neutralising antibody to serotypes 2 and 4 [93]. Surprisingly, infants infected with serotype 1 strains experienced a neutralising antibody boost to human serotypes 1–4 [93]. This may probably explain the apparent reduction in the incidence of all-cause diarrhoea mortality in <1-year vs. <5-years-old children between 2006 and 2009 in four South American countries without RoV implementation—Argentina (32.5% vs. 33.8%), Chile (30.9% vs. 65.7%), Costa Rica (61.6% vs. 50.6%), and Paraguay (51.2% vs. 50.8%). These protections conferred, probably by natural infection by wildtype strains, were highly competitive with vaccine-acquired protection in five Latin American countries (Brazil, El Salvador, Mexico, Nicaragua, and Panama), which reported a 30–50% rate reduction in diarrhoea after the introduction of RoVs in 2006 [177]. However, De Oliveira and co-workers showed a reduction in the trends of all-cause diarrhoea-related deaths and hospitalisations in children <5 years from Bolivia, El Salvador, Honduras, and Venezuela over two–four years of post-vaccination. Relatively, there was no reduction observed in Argentina over the same period, probably because there was no national rotavirus vaccination yet [178]. This suggests, perhaps, the protection offered by the natural infection is ephemeral.

Early exposure to RVI is likely common in Africa than other continents, and this can affect vaccine-acquired protection at the time the first dose is administered. This can be attributed to the high level of pre-vaccination IgA/seropositivity baseline in these children. The occurrence of any RVGE in Africa was higher than in Europe during the first year of life (≤2.78% vs. ≤2.03% per month) but much lower during the second year (≤0.86% vs. ≤2.00% per month) [377], which probably suggests partial protection by multiple natural infections in African children. Cunliffe and co-workers observed a higher rate of anti-rotavirus IgA seropositivity before the first dose (6.4–12.2 weeks) and after the second dose of Rotarix^®^ (13.5–19.6 weeks) in Africa, Asia, and Latin America, with pre-D1 seropositivity of 2.1–26.3% and post-D2 of 6.3–34.8%, than the developed countries of Asia, Europe, and North America, with a pre-D1 of 0–9.4% and post-D2 of 0–21.3%, respectively [378]. This observation suggests early exposure to natural RVI before receiving the first dose of RoV. Several longitudinal studies have reported natural protection offered by multiple neonatal RVIs. Successive naturally occurring RVIs offered a complete homotypic and heterotypic protection against asymptomatic and symptomatic moderate to severe gastroenteritis in subsequent re-infection in Mexico [122].

Similarly, Bernstein and co-workers reported protections offered by asymptomatic and symptomatic primary infections, and that a correlation existed between the age and reduction in the ratio of symptomatic to asymptomatic primary infections [376]. This technically means that the higher the age, the lesser the occasion of symptomatic RVIs. In Native American infants, a 72% overall preventive effectiveness was offered by the primary symptomatic RVI, and the initial episode of infection caused by the serotype G3 provided homotypic protection of 91% against subsequent G3 episodes [379]. Lastly, Ward and co-workers reported about a 93% overall efficacy against asymptomatic RVI following natural RVI in a large two-year placebo recipient during a vaccine trial in the USA [380].

Protection from multiple natural infections has been reported against subsequent symptomatic RVI in Guinea-Bissau [381], and this was largely assigned to the protective role of maternal transplacental-acquired IgG and secretory IgA from breastfeeding [382,383,384]. In Nicaragua, there was a high correlation between the first demonstration of rotavirus in stool samples and the concentration of colostrum secretory anti-rotavirus IgA antibodies [384]. Further, in Egypt, a lower incidence of RVD was associated with breastfeeding in infants <12 months but unlikely connected to water quality and sanitation [385]. To a lesser extent, therefore, naturally acquired or serotype-specific immunity only conferred partial protection against the re-occurrence of RVD [386].

In India, multiple natural infections have been shown to offer protection [387]. Compared with Mexico and Guinea-Bissau, the immune response offered in Indian children was less protective and lacked homotypic/heterotypic protection. The reason is not far-fetched. There were much-reported cases of earlier occurrence of natural RVIs in Indian infants, which may interfere negatively with the maturation of their immune systems—an evocative of the overwhelming juvenile immune system of developing infants. In fact, Bhan and co-workers reported about 60% of asymptomatic infections in the cohort of the newborn in New Delhi as early as the fourth day of life and this was predominantly caused by an unusual G9P[11]. This G9P[11] strain was later argued to be considered as a vaccine strain and was studied further to see whether it can offer natural protection [388]. In Australia, neonatal RVI seemed not to offer protection against re-infection but protected against the development of clinically severe symptoms during re-infection [99].

In all, the globally common genotypes that cause primary neonatal infections were G1P[8], G2P[4], G3P[8], and G4P[8] [389,390], and in Africa, G1P[8], G2P[4], G2P[6], G3P[4], G3P[6], G3P[9], G3P[8], G4P[8], G4P[6], G8P[1], G8P[4], G8P[6], and G9P[6] were the most common typable genotypes [391,392]. In India, G1P[8], G2P[4], G3P[8], and G4P[8] were common as those found globally; however, strains like G1P[6], G2P[6], G3P[6], G4P[6], and G9P[6] were unique and primarily found in asymptomatic neonates but were rare in children with diarrhoea [393,394]. In addition to common genotypes, G1P[8], G2P[4], G3P[8], and G4P[8], and some uncommon serotypes of G1P[3], G1P[6], G1P[9], G2P[6], G3P[3], G3P[6], G3P[9], G4P[6], and G5P[8] represent one-third of all rotavirus-related diarrhoea cases in Brazil [395,396], and this may represent the circulating serotypes in the South American countries.

The occurrence of mixed infections from strains of different genetic subgroups representing the natural reassortment has been reported. Fischer and co-workers reported co-infection of G1P[4]/[6] and G2P[4]/[6] genotypes, which represented two different regional rotavirus wildtype strains engaging in natural reassortment [392]. Mphahlele and Steele also reported some dual infections between P[4] and P[8] genotypes over ten years of the relative frequency of human rotavirus VP4 (P-genotype) in South Africa [397].

This diversity in the rotavirus genotypes and year-to-year variation may be of significant issues to public health regarding vaccine formulation and regional-specific protection. The low VE in a few areas, which had been associated with the emerging rotavirus antigenic variants produced by the escaped vaccine mutants, is another pressing issue. Therefore, global surveillance over several years and an adequate understanding of the shift in the genotype distribution will further encourage suitable live-attenuated rotavirus vaccine strains or upgrading of the current vaccines for effective vaccine-acquired protection against RVI.

## 10. Conclusions, Recommendation, and Future Direction

There is no gainsaying that RoVs are effective against all forms of RVIs. The use of these vaccines has brought a dramatic reduction in the global epidemics of diarrhoea in infants, toddlers, and children up to 5 years of age and even beyond. Reports from health workers have confirmed a drastic reduction in the rate of hospitalisation due to diarrhoea and RVI. The laboratory detection rate for RVI is constantly reducing annually with sustainable reductions. The impacts of these LAORoVs, especially Rotarix^®^ and RotaTeq^®^, are huge, particularly in the LMICs where the infection rate is high. The efficacy of these vaccines is so high in the HICs to the extent that RVI is almost completely under control unlike the low resource countries in Africa and Asia, where the infection rate is still high, but the vaccine efficacy is relatively low. With this low vaccine efficacy, however, more lives have been saved than in the high resource countries. The reasons for lower vaccine efficacy in the LMICs are constantly being studied.

Vaccines remain the only pragmatic approach to curb childhood RVIs. The increasing acceptability and implementation of RoVs in recent times are highly encouraging. Africa has the highest vaccine implementation, while Southeast Asia has the lowest [35]. GAVI has continued to support vaccine implementation in most of the low resource countries. Unrelentingly, WHO, GAVI, United Nations Children’s Fund (UNICEF), PAHO, and PATH have continued to recommend and support the implementation of these vaccines, especially in the LMICs where the infection rate is greatly felt. The national recommendation for rotavirus vaccination has been issued in many regions globally, for example, the US Advisory Committee on Immunization Practices (ACIP), WHO for the global recommendation, the European Society for Paediatric Infectious Diseases (ESPID), and the European Society for Paediatric, Gastroenterology, Hepatology and Nutrition (ESPGHAN) for Europe [128]. All supported and recommended rotavirus vaccination worldwide against RVGE. Amid this, WHO licensed two more Indian-made LAORoV (Rotavac^®^ and RotaSIIL^®^) to buttress the seemingly overwhelming production of the previously licensed vaccines (Rotarix^®^ and RotaTeq^®^). To further curb this infection, countries like China and Vietnam developed regional LAORoVs, while other countries such as Australia, Brazil, Indonesia, South Africa, the UK, the USA, and Venezuela are fast coming up with more and diverse polyvalent LAORoV and sub-units vaccine candidates [125]. In addition to this, non-replicating rotavirus vaccines such as VP6-DNA, inactivated rotavirus particles, sub-unit capsid proteins, and RV-VLPs are becoming potential alternative candidates against RVI, and their trials are very convincing [38,125,188].

Paradoxically, RoVs are cost-effective yet relatively more expensive than most of the children routine vaccines. With the help of GAVI, eligible countries can now co-finance with as low as a two-third price reduction, while GAVI procures at USD 2.08/dose of Rotarix^®^ and USD 3.20/dose of RotaTeq^®^ through UNICEF from 2017 to 2021 [398,399]. With Rotavac^®^ and RotaSIIL^®^, the GAVI purchasing price is as low as USD 1/dose and USD 2/dose, respectively [55]. Unfortunately, the current global COVID-19 pandemic, which will leave a lasting imprint on the global economy let alone individual countries, will definitely affect the implementation of RoVs in the remaining countries that are yet to introduce these vaccines. Not only this, but continuity from other countries that have introduced these vaccines may also be dicey. This is because RoVs are relatively more expensive than any other childhood routine vaccine, and it is the single infant vaccine mostly subsidised by third-party non-governmental organisations (NGO). Many countries are already in financial debt to curb the COVID-19 pandemic, and this may leave them with an option to forestall the RoV procurement. Already, CDC has reported a “notable decrease” in the number of vaccines ordered through the federal program for childhood vaccination since the beginning of the SARS-CoV-2 pandemic in the USA [400]. Notwithstanding, the evidence of vaccine effectiveness surrounding the introduction and implementation of RoVs and other routine childhood vaccines, together with their accrued efficacy, are substantial enough to convince the policy- and decision-makers to ensure the continuous procurement of these vital and life-saving children vaccines.

## Figures and Tables

**Figure 1 vaccines-08-00341-f001:**
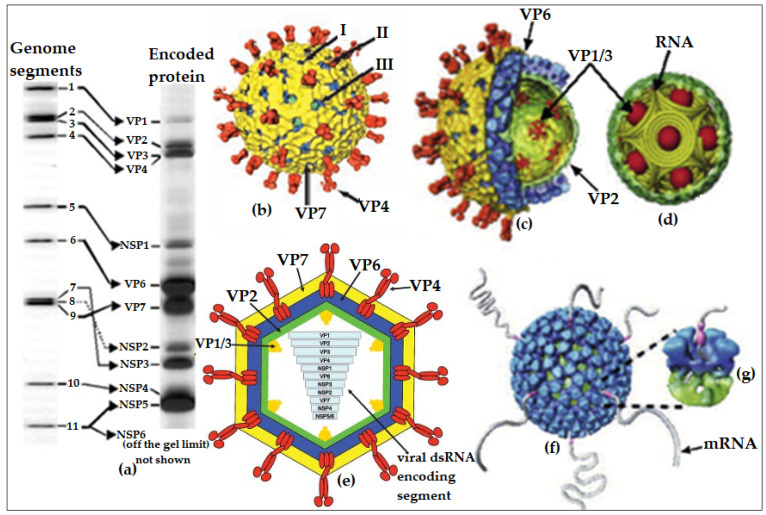
Detail illustration of rotavirus particle and its segmented dsRNA genome encoding protein: (**a**) Electrophoretic pattern of rotavirus group A (RVA) individual segmented genomic RNA with their corresponding encoded proteins. The proximity of segments 7, 8, and 9 represents the identity of rotavirus electrophoretic pattern; (**b**) cryo-electron microscopy reconstruction (CEMR) of the rotavirus triple-layered particles (TLPs) with the spike-like protein VP4 and the outer-layer glycoprotein VP7 (shaded in yellow) with specific localisation of transcriptional pores—I, II, and III; (**c**) a semi-longitudinal section of TLPs showing the inner capsid (VP6) and core protein (VP2) layers and the transcriptional enzymes accessories (VP1 and VP3) enclosed by the VP2 core protein; (**d**) genomic organisation in rotavirus core protein (VP2) with intricate transcriptional enzymes enclosed inside the protein coat; (**e**) complete longitudinal section of rotavirus TLPs showing the chronological arrangement of the segmented genomes (VP1 to NSP5/6); (**f**,**g**) model from CEMR of transcribing double-layered particles (DLPs) showing the endogenous transcription, which results in the simultaneous release of the transcribed mRNAs. Figures adapted and modified from [12,13] with permissions.

**Figure 2 vaccines-08-00341-f002:**
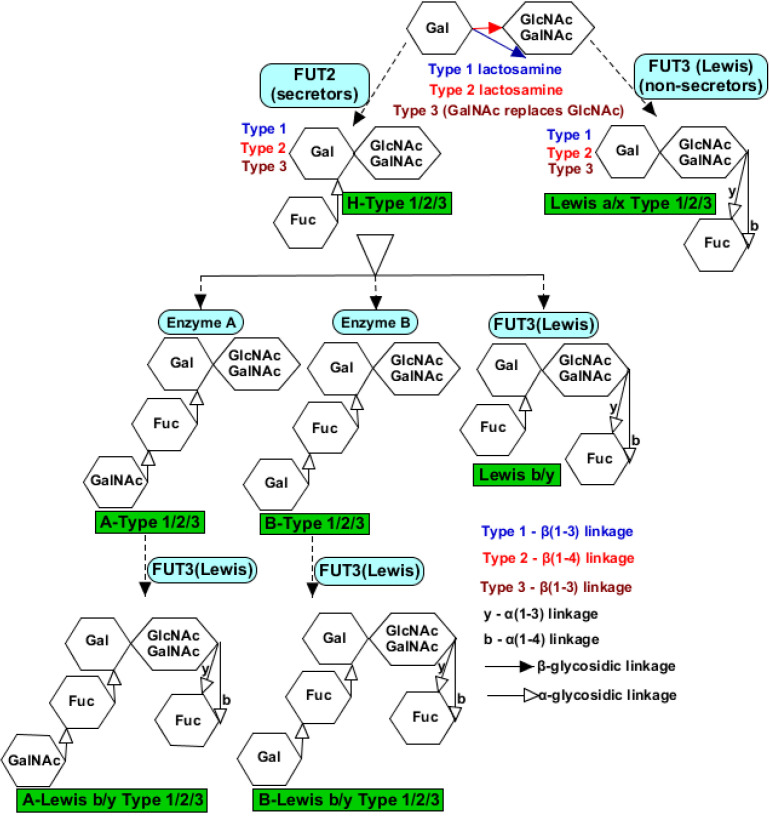
Terminal oligosaccharides of histo-blood group antigens (HBGAs) receptors that facilitate rotavirus VP8* binding. Terminal fucose is catalysed by α-1,2-fucosyltransferase 2 encoded by the *FUT2* gene, generally referred to as “secretor” to form an H-type glycol-receptor, which can remain unmodified as type O^+^ secretor. However, when terminal fucose is catalysed by α-1,4-fucosyltransferase 3 encoded by the *FUT3* gene and remained unmodified, it becomes type O^-^ non-secretor. Subsequent modification produced A-type, B-type, Lewis, A-Lewis, and B-Lewis. Terminal -GalNAc in the A-type makes it vulnerable to many rotavirus strains and enhances inter-species transmission.

**Table 1 vaccines-08-00341-t001:** Development and pre-clinical assessments of notable live-attenuated oral rotavirus vaccines.

Vaccine	Strain	Developer/Manufacturer	Type of Vaccine/Status	Vaccine Development and Assessment
**From animal rotavirus strains (first generation of rotavirus vaccines)**				
RIT4237 (monovalent)	G6P6[1], BRV NCDV strain	SmithKline-RIT (Rixensart, Belgium)	Jennerian, but withdrawn	About 154 cell culture passages with high titre value of over 10^8^ TCID_50_/mL, first to be tested in human, higher efficacy in the developed countries, negative gastric effect, breast milk did not affect but suppressed by OPV [154].
RRV-MMU (monovalent)	G3P[5] RRV strain	National Institute of Health (NIH)	Modified Jennerian-based vaccine, but withdrawn	Passaged 9 and 7 times in monkey kidney and foetal rhesus lung cells, respectively, with 10^5^ PFU/dose titre level, reactogenic for fever, very virulent and later used as a backbone for RotaShield^®^ [81,155].
WC3 (monovalent)	Wistar calf strain G6P[7]	Institut Merieux/Wistar Institute	Jennerian, but withdrawn	Passaged 20 times with 10^7^ PFU/mL titre value, less efficacious compared with RRV-MMU but later used as a backbone for RotaTeq^®^ [91,156].
LLR-37 (monovalent)	Lanzhou lamb-derived rotavirus G10P[15]	Lanzhou Institute of Biological Products (China)	Jennerian, licensed for use in China only since the year 2000	Ovine attenuated vaccine, efficacious in 2 doses at 2 months and 2 years, replicate with faecal shedding within 15 days post-vaccination. Over 60 million doses rolled out as at the end of 2014 [157,158].
**Animal–human reassortant strains (second generation of rotavirus vaccines)—Jennerian-based vaccines**				
RotaShield^®^ (RV4)	Tetravalent human–rhesus reassortant (G1-G4)P7 [5]	Wyeth (Madison, NJ, USA)	Jennerian and first licensed to be used in the USA by 1998 but withdrawn in 1999 due to IS, but now in phase II trials for neonatal administration in Ghana	First reassortant prototype, stable at 25 °C, formulated with G1, G2, and G4 VP7 antigens on rhesus G3P[5] backbone, reactogenic for fever, formulated with 4 × 10^4^ PFU/mL with two doses and caused IS but tested to be safe in neonates in Ghana [159], which might favour its re-introduction [63,160].
RotaTeq^®^ (RV5)	Pentavalent human–bovine reassortant G1-G4+P7 [5]; G6+P1A [8]	Merck & Co., Inc. (Burlington, MA, USA). Presented as a 2.5 mL liquid vaccine suspended in a buffer and administered via a squeeze tube. Scheduled as 3Ds	Jennerian, pre-qualified for global use since 2008 but first licensed in 2006. Usually co-administered with DPT 1, 2, and 3 vaccine doses	It is formulated with 2–2.8 × 10^6^ PFU/mL with immunogenicity tested in 70,000 infants [161]. The overall efficacy was 95% against (G1–G4, G9)P[8] and displayed heterotypic protection against G2P[4]. Shedding and diarrhoea were observed in <1% infants [162,163]. Similarly, high efficacy was reported in 34,035 vaccinated infants from 11 countries across Latin America, America, and Europe [123]. VE increased with severity but decreased over 2 years of follow-up in Africa and Asia with 6674 infants from Ghana, Kenya, Mali, Bangladesh, and Vietnam (4705 African and 1969 Asian) [164]. The vaccine, however, requires cold-chain (2–8 °C) storage and transportation.
BRV-TV or BRV-PV	Bovine–human tetravalent reassortant G1-G4+P7 [5]; G1-G4+G9-P7 [5]	Biotecnics Instituto Butantan Shantha Biotecnics Limited Serum Institute of India (Pune, India) Wuhan (China)	Phase I trial in Brazil Phase II/III trials in India Phase III trial in China -	Started by Wyeth but discontinued at the same time RotaShield^®^ was withdrawn; however, the prototype has been licensed by NIH to Brazil, India, and China [165].
RotaSIIL^®^ (BRV-PV)	Pentavalent human–bovine reassortant serotypes G(1–4) and G9	Serum Institute of India Pvt Ltd. (SIIPL). Presented in two forms–2.5 mL reconstituted lyophilised vial and 2.5 mL prepared liquid vial. Each scheduled as 3Ds	Pre-qualified for global use in 2018 but licensed for use in India since 2016. Co-administered with DPT 1, 2, and 3 vaccine doses	World’s first cost-effective, thermostable vaccine which can be stored without refrigeration or below 25 °C. It is a bovine UK rotavirus strain from NIH (USA), which was reassorted with human VP7 genome encoding segments [166]. The vaccine has a lifespan of 20 months at 37 °C or 7 months at 45 °C.
**Human rotavirus strains (second generation of rotavirus vaccines)—modified Jennerian-based vaccines**				
Rotarix^®^ (RIX-4414)	Monovalent human strain G1P1A [8]	GlaxoSmithKline, GSK (Belgium). Presented in two forms. First, as 1 mL liquid vaccine with oral applicator or as a squeeze tube. Secondly, as 1.5 mL freeze-dried reconstituted with buffer and used with an oral applicator. Scheduled as 2Ds	Pre-qualified for global use since 2009 but first licensed in 2006. Co-administered with DPT 1, 2, and 3 vaccine doses	The most extensively used vaccine that was initially developed in Cincinnati, USA. The parent strain was first isolated from a child with natural RVI during the serotype 1 outbreak in 1988–1989 [91] and passaged 45 times in MRC-5 cells to remove the infectivity. Initially developed by Avant Immunotherapeutics Inc. Massachusetts, USA but licensed to GSK for further modification [167]. With 60,000 infants, Rotarix^®^ induced >90% efficacy with significant homotypic and heterotypic protection against HRV strains, reaching 100% efficacy against a more severe case [127]. Summarily, Phases I–III trials with over 70,000 children proved the vaccine was non-reactogenic, well-tolerated, effective at first dose, but increased seropositivity rate after the second dose, not associated with IS but provided >85% protection against moderate to SRVGE, and reduced GE-related hospitalisation by >40% in Latin America and >75% in Europe [168,169]. This vaccine also requires cold-chain facilities (2–8 °C).
Rotavac^®^	Human neonatal strain 116E G9P[11]	Bharat Biotech International Limited (India). Presented as a 0.5 mL liquid vaccine with a separate dropper	Pre-qualified for global use in 2018 but licensed for use in India since 2014 [170]. Co-administered with DPT 1, 2, and 3 vaccine doses	It is a naturally occurring reassortant strain G9P[11], containing one BRV gene P[11] and ten HRV genes [60] at 10^5^ FFU/dose formulation, taking in 3 single doses at 4 weeks interval starting from 6 weeks of age. This vaccine was licensed and introduced into the NIP (National Immunisation Program) of India by 2014 [60]. This vaccine must be stored at −20 °C.
Rotavin-M1^®^	Human strain G1P1A [8]	Centre for Research and Production of Vaccines and Biologicals - Polyvac (Vietnam)	Licensed to be used in Vietnam since 2007	It was derived from an attenuated G1P[8] strain (KH0118-2003) isolated from a Vietnamese infant, passaged in Vero cell at 10^6.3^ FFU/dose, given in 2 doses (2 months apart) or 3 doses (1 month apart), safety, and immunogenicity tested in 160 infants with rotavirus-specific IgA seroconversion rate of 73% vs. 58% in Rotarix^®^ and rotavirus shedding between 44% and 48% vs. 65% for Rotarix^®^. Over 500,000 doses already rolled out. The vaccine must be stored at −20 °C [19].
RV3-BB	Human neonatal strain G3P2A [6]	Murdoch Children Research Institute, Australia Biofarma, Indonesia	Phase III trial in Indonesia	RV3 is based on a neonatal G3P[6] strain from Melbourne. A Vero cell-adapted derivative of the strain RV3-BB grew to a higher titre 8.3 × 10^6^ FFU/mL and was more immunogenic than the parent strain RV3 [171]. The Phase IIa trial with 3Ds in New Zealand showed 63% neonate and 74% infant with IgA seroconversion and shedding was 70% in the neonate and 78% in the infant. The vaccine was well-tolerated and immunogenic [172]. Analysis in the Phase IIb in Indonesia showed the efficacy of 75%, 51%, and 63% in the neonate, infant, and combined groups against SRVGE, respectively [173]. Vaccine response/“take” of RV3-BB was 94% and 99% in the neonatal- and infant-scheduled groups, respectively [173].

D = dose; DPT = Diphtheria, Pertussis and Tetanus; IS = intussusception; NIH = National Institute of Health; SRVGE = severe rotavirus gastroenteritis; IgA = immunoglobulin A; FFU = focus-forming unit.

**Table 2 vaccines-08-00341-t002:** Comparative analysis of rotavirus vaccine efficacy and effectiveness in pre-clinical trials.

Vaccine	Seasonal Efficacy/Follow-up/Doses		Remark
	First-Year Follow-up	Second-Year Follow-up	
North America and Europe			
Rotarix^®^ was tested against RVGE in infants from six different European countries	87.1% efficacy against any SRVGE and 75–96% efficacy against the corresponding genotypes G1–G4 and G9	71.9% efficacy against any SRVGE and 69–84% efficacy against the corresponding G1–G4 and G9. About 85.6% efficacy against SRVGE and 77–97% efficacy against the corresponding G1–G4 and G9	Observed combined efficacy against any SRVGE and SRVGE was 78.9% and 90.4% respectively. The combined efficacy against all-cause SRVGE and hospital admission was 49.6% and 71.5%, respectively [135]. Though similar reactogenic was observed in the tested groups, serious and nonserious adverse effects were reported, but no IS case in the USA and Canada [168].
Rotarix^®^ was evaluated for safety and immunogenicity	95.8% efficacy against SRVGE and 74–100% against the G1–G4 and G9		2Ds of RV1 are well tolerated with a high vaccine “take”, highly immunogenic to induce IgA without side effect or interference with routine childhood vaccines [174].
RotaTeq^®^ evaluated for efficacy and safety in >30,000 infants out of 70,000 infants’ enrolment for REST analysis in 5 European countries	100% efficacy against SRVGE and 72% efficacy against any RVGE	94.3% efficacy against SRVGE and 58.5% efficacy against any RVGE	RotaTeq^®^ showed the efficacy of 98.3% and 68% respectively against SRVGE and all-cause of RVGE due to any serotype for the two rotavirus seasons in Europe. Further, 94.5% efficacy against RVGE hospitalisation and EV due to any serotype over 2 years of vaccination without any significant safety concern [121].
**Latin America**			
Rotarix^®^ trial in 63,225 infants in 1-year Phase III trial from 11 Latin American and Finland	FDs produced 85% efficacy against SRVGE and hospitalisation, reaching 100% against VSRVGE with 45% efficacy in reducing any-cause of rotavirus diarrhoea hospitalisation		Two oral doses of the live attenuated G1P[8] HRV vaccine protected infants without associating with an increased risk of IS [127].
Rotarix^®^ effectiveness in Bolivia	RV1 offered sustainable protection through two years of the life of Bolivian children, with similar VE against hospital admission <1 year and >1 year of age		RV1 offered significant protection against hospital admission for diarrhoea caused by diverse serotypes: G9P[8], G3P[8], G2P[4], and G9P[6], which are either partially and fully heterotypic to the G1P[8] vaccine [175]. This protection was sustained over two years of life against diverse serotypes different from the vaccine strain.
Rotarix^®^ introduction vs. reduction of all-cause diarrhoea death in Brazil, El Salvador, Mexico, Nicaragua, Panama, Venezuela, Bolivia, and Honduras	There was a significant reduction in all-cause diarrhoea mortality in the post-vaccination period among children <5 years old between 2006 and 2009 compared with the pre-vaccination period [176]. This reduction, 30–50%, was significant in all the vaccine adopter countries except Nicaragua. However, non-vaccine adopter countries (Argentina, Chile, Costa Rica, and Paraguay) showed a greater mortality reduction in the pre-vaccination period, 2002–2005 [176]		Significant reduction after post-vaccination in these countries was observed except in Panama, where this reduction was not observed. Concomitantly, countries such as Argentina, Chile, Costa Rica, and Paraguay were not using the vaccine at the time of this assessment [176]. Vaccine impact on the reduction of deaths was more than the hospitalisation [177].
Duration of protection offered by RotaTeq^®^ in Nicaragua	Risk of rotavirus hospitalisation was two-fold lower in vaccinated children <1 year compared with ≥1 year	There was no significant reduction in rotavirus hospitalisation in the second year after the last vaccine dose	Waning immunity was suspected, and a booster dose was suggested in Nicaragua [178].
RotaTeq^®^ introduction vs. reduction of all-cause diarrhoea death rates in Mexico and Nicaragua	Significant reduction in all-cause diarrhoea mortality in <5 years between 2006 and 2009 when comparing postvaccination and pre-vaccination together		A significant reduction was observed in Mexico, but not in Nicaragua [176].
**Africa**			
Rotarix^®^ efficacy against severe diarrhoea	After the first year of follow-up, 61.2%, 58.7%, and 63.7% overall efficacy against SRVGE were observed in South Africa and Malawi combined. In South Africa, the efficacies of 2 and 3Ds were 72.2% and 81.5% and a pooled of 76.9%, while in Malawi, 49.2% and 49.7% with a pooled of 49.4% were respectively observed. Efficacy against all-cause SRVGE was 30.2%, and when pooled, South Africa was 44.1%, while Malawi was 25.1% [130]	There was a marginal reduction in vaccine efficacy in the second year of follow-up in Malawi. The 2Ds, 3Ds, and pooled efficacies against SRVGE were 2.6%, 33.1%, and 17.6% while against severe any-cause RVGE were 13.2%, 5.2%, and 9.3% respectively. There was 9.3% efficacy against all SRVGE, while the entire follow-up yielded 15.9% vaccine efficacy [179]	RV1 significantly reduced the incidence of RVGE among African infants during the first year of life. No difference in vaccine efficacy against G1 and non-G1 (G2, G8, G12) [180,181]. Madhi and co-workers observed a higher preventable episode of SRVGE/100 infants vaccinated/year in Malawi (6.7) than South Africa (4.2), though vaccine efficacy was lower in the former than the latter [130]. From Malawi, RV1 significantly reduced the incidence of SRVGE caused by diverse rotavirus strains and more episodes of SRVGE, and any SRVGE seemed to be prevented at 2Ds more than 3Ds, even after the entire follow-up [179]. Vaccine-acquired immunity seemed not to endure beyond the first year of vaccination. Three doses boost vs. 2Ds of RV1 in Malawi at the first year of follow-up produced 27.2% vs. 22.9% efficacies against all-cause SRVGE, while after the second year of follow-up, 5.2% vs. 13.2% efficacies were observed and the entire follow-up yielded 15.7% vs. 16% efficacies [179].
RotaTeq^®^ efficacy between 2007 and 2009	Prevention of SRVGE in Ghana was 65%, Kenya 83.4%, Mali 1%, and overall was 64.2%	Prevention of SRVGE in Ghana was 29.4%, Kenya 54.7%, Mali 19.2% and overall was 19.6%	Overall efficacies of 30.5%, 39.3%, 40.7%, and 100% were observed against any severity, moderate-to-severe, severe, VSRVGE, respectively. GE was the most common adverse event associated with this vaccine, and this occurred within 14 days of any dose. There was no significant difference in this adverse events rate (0.6% each) between the vaccinated and the placebo. High IgA sero-response rates of 78.9%, 73.8%, and 82.5% were observed in Ghana, Kenya, and Mali, respectively, but very low serotype-response rates to G1–G4 and P1A[8]. Overall efficacy against moderate-to-severe RVGE was 55.5%, 63.9%, and 17.6% in Ghana, Kenya, and Mali, respectively. Generally, the efficacy of RV5 in sub-Sahara Africa was higher with higher severe cases of RVD prevented/100 vaccinated/year in the first year of follow-up than the second year, however, because of the intense mortality rates in Africa, the magnitude of prevented cases is very high than any other region [136]. Note: Infants with HIV infection were included in this study
RotaSIIL^®^ efficacy assessed in 3508 infants from Niger	In a 2-stratum treatment (ITTP vs. PPP), the efficacy of 3Ds of BRV-PV against all RVGE (33% vs. 34.5%), against SRVGE (69.1% vs. 66.7%), against VSRVGE (77.4% vs. 78.8%), against all GE (−12.4% vs. −0.6%) against SGE (16.2% vs. 10.7%), and against VSGE (67.7% vs. 68%). Cases of SRVGE/100 infant vaccinated/year in the vaccine vs. placebo groups were 2.14 vs. 6.44		At 66.7% efficacy, BRV-PV prevented 4.3 episodes of SRVGE/100 infants vaccinated/year. No significant difference in the adverse events and mortality rate. No IS cases. Efficacy increases with clinical severity. This vaccine offered protection against SRVGE resulting in mortality and morbidity of infants, and there is no need for cold storage facility (stable for 2 years at 37 °C or 6 months at 40 °C) [182].
Southeast Asia			
RotaTeq^®^ efficacy in 2036 recruited infants from Bangladesh and Vietnam	Overall efficacy against SRVGE in Bangladesh was 42.7% with the first-year follow-up of 45.7% while Vietnam was 63.9% with the first-year follow-up of 72.3%	The second-year follow-up for Bangladesh was 39.3%, and Vietnam was 64.6%	Overall efficacy of 42.5%, 48.3%, and 70% against any severity, SRVGE, and VSRVGE, respectively, for nearly 2 years of follow-up was reported. Vaccine efficacy increases with clinical severity. Higher efficacy was consistently found in Vietnam than in Bangladesh. However, efficacy against the VSRVGE (Vesikari score ≥ 15) was higher in Bangladesh because, before the vaccination, the infant mortality rate in Bangladesh (29.7/1000 live births) was twice Vietnam’s (14.7/1000 live births), and again the SRVGE cases were substantially higher in Bangladesh than Vietnam [57]. Pneumonia was one of the serious adverse events, which occurred within 14 days of any dose, and there was no significant difference in this adverse event between the vaccinated and the placebo [57].
**India**			
Rotavac^®^ assessed for efficacy in India with 4532 infants ranging from 6 to 8-weeks-old	Efficacy of 56.4% against SRVGE with an incidence rate of 1.5 in vaccine vs. 3.2 in placebo groups. Six intussusception and 25 deaths in vaccine vs. 2 intussusception and 17 deaths in placebo groups were observed after the 3rd dose	Efficacy of 55.1% against SRVGE with an incidence rate of 1.3 in vaccine vs. 2.9 in placebo groups. Eight intussusception and 30 deaths in vaccine vs. 3 intussusception and 18 deaths in placebo groups were observed in the second year of follow-up	Whether from ITTP or PPP analysis, Rotavac^®^ (116E) was effective and well-tolerated in Indian infants and maintained this efficacy in the second year after the 3Ds vaccination program. The reported adverse events were the same in the two groups—vaccine vs. placebo. IS occurred after 112 vs. 36 days at 1st year and 112–587 vs. 36–60 days at 2nd year, respectively. Increased seroconversion 4 weeks after the 3rd dose was observed, which suggested wildtype RVI in infants before the vaccination period [183]
RotaSIIL^®^ assessed for efficacy in Phase III clinical trial in India with 3749 infants ranging from 6 to 8-weeks-old	Efficacy against VSRVGE was 60.5% with PPP analysis or 61.3% with ITTP analysis	Efficacy against VSRVGE was 54.7% with PPP analysis or 52.9% with ITTP analysis	Reactogenic response compared to placebo after 1st dose showed mild irritability (42.6% vs. 36.1%), severe fever ≥39.5–<40.5 °C (4 vs. 1 case), and LRTI (60 vs. 38 cases). All other adverse events were the same in the two arms. Overall reduction of SRVGE incidence was 1.7 cases/100 person/year in the ITTP. Generally low serum anti-rotavirus IgA seroresponses (33.6% vs. 9.7%), however, BRV-PV was well tolerated and a safe vaccine in Indian infants, but the efficacy is very low especially against SRVGE [61]. There was no vaccine interference with OPV because similar seroconversion to OPV was observed in the two arms.

BRV-PV = bovine rotavirus pentavalent vaccine; D = dose; EV = emergency visit; FD = full doses; ITTP = intention-to-treat population; LRTI = lower respiratory tract infection; OPV = oral polio vaccine; PPP = per-protocol population; REST = rotavirus efficacy and safety trial; RoV = rotavirus vaccines; RVI = rotavirus infection; RVGE = rotavirus gastroenteritis; SRVD = severe rotavirus diarrhoea; SRVGE = severe rotavirus gastroenteritis; VSRVGE = very severe rotavirus gastroenteritis; VE = vaccine effectiveness. By the end of 2019, 80 countries (47 from GAVI-eligible countries) are using Rotarix^®^, 15 (4 from GAVI-eligible countries) RotaTeq^®^, 9 countries use both vaccines, and probably 3 countries with their national rotavirus vaccines making 98 countries (46 from GAVI-eligible countries). However, only 62 countries (29 from GAVI-eligible countries) have an available report on the RoV evaluation [141]. Summarily, the vaccine efficacy against SRVGE in the USA and Europe is 91%, in South America 80%, in Asia 50%, and in sub-Sahara Africa 46%. Correspondingly, the efficacy against all forms of diarrhoea-associated hospitalisation due to rotavirus is 94% in the USA and Europe, 84% in South America, 94% in Asia, and 58% in sub-Saharan Africa [126,184].

**Table 3 vaccines-08-00341-t003:** Comparison between RV3-BB and RIX4414 neonatal oral live-attenuated human rotavirus (HRV) candidates.

Dose	*RV3-BB*				*RIX4414 ≈* Rotarix^®^			
	IgA Response (Neonates)	Neonatal Stool Shedding	IgA Response (Infants)	Infant Stool Shedding	Vaccine Concentration (FFU/mL)	IgA Response (Infants)	Infant Stool Shedding	
							1D	2Ds
1	23%	4%	Placebo	-	10^4.7^ (with antacid)	100%	42%	8%
2	54%	42%	44%	41%	10^4.1^ (with buffer)	74%	38%	13%
3	94%	70%	86%	62%	10^4.7^ (with buffer)	93%	60%	8%
4	Placebo	-	99%	76%	10^5.8^ (with buffer)	96%	55%	0

Shedding of RV3-BB was 3 and 7 days after vaccine administration while shedding of RIX4414 was 7 and 9 days after vaccine administration.

**Table 4 vaccines-08-00341-t004:** Host range preferences for the rotavirus P genotypes and HBGAs glycol-receptors.

P Genogroup	P Genotype	Animals				Human				HBGAs Preference
P[I]	P[1]									Sialic acid-dependent
	P[2]									
	P[3]									
	P[5]									
	P[7]									
	P[10]									
	P[12]–[13]									
	P[15]–[16]									
	P[18]									
	P[20]–[24]									
	P[26]–[29]									
	P[32]–[34]									
P[II]	P[4]									Type 1 precursors, H-type 1 and Lewis ^b^ P[6] binds only H-type 1
	P[6]									
	P[8]									
	P[19]									
P[III]	P[9]									A-type antigen
	P[14]									
	P[25]									
P[IV]	P[11]									Type 2 precursors
P[V]	P[17]									-
	P[30]–[31]									
	P[35]									

Colour intensity shows the preference infected host by the corresponding genotype. The darker green indicates a strong host preference while the lighter green indicates a weak host preference for each of the corresponding genotypes.

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
