# Peer review of "Overview of the Development, Impacts, and Challenges of Live-Attenuated Oral Rotavirus Vaccines"

_vaccines, 2020, doi:10.3390/vaccines8030341_

Round 1
Reviewer 1 Report
The review needs considerable editing down. Its starts off really interesting. However I got to the paragraph starting at 234 and made notes that the content started to drift and probably should be put in a table. I kept reading and really the review lost focus. I then looked at the number of pages in the PDF and was shocked.
I doubt very my much that anyone will read this volume of information. Certainly not in this journal. This is highly specialised content.
I am sure the content is fine but there is no way I can verify everything.
I apologies if I have caused offense but for the reader of this journal this is not appropriate.
The content is more appropriate for a book.
Author Response
Dear Editors,
Please, see the attachment for the response to reviewer 1 comments.
Thanks

Reviewer 2 Report
In this manuscript, the authors reviewed the development, production and efficacy and future challenges of live attenuated rotavirus vaccine. Initially, the authors introduced rotavirus particles followed by a discussion about available rotavirus vaccines, their implementation methods, efficacy, cost and potential challenges of these vaccines. Then they elaborated on the early attempts to produce live-attenuated oral rotavirus vaccines and the potential success of these efforts. The authors provided comparative account of vaccine coverage, effectiveness, and efficacy and the factors that account for the efficiency of the rotavirus vaccine. The advent of these vaccines also caused vaccine-induced selective pressure of different strains. Here authors discussed the impact of rotavirus vaccines on the genotype distribution pattern in general population. Then they discussed the role of naturally acquired protection against rotavirus infections. Finally, the authors summarized the review provided recommendations based on their conclusions.
Overall the manuscript is well structured and written in clear language. However, some of the sections are disproportionately long.
Section 3 ‘Initial attempts to produce live-attenuated oral rotavirus vaccines’ can be shortened significantly by summarizing some of the earlier studies.
Section 10. “Conclusion, recommendation, and future direction’ should be shortened by removing redundant text mentioned elsewhere. In this section, the authors should provide a crisp take-home message to the readers. The authors may consider rewriting this section to highlight the critical messages in the review.
The authors used many abbreviations throughout the text and sometimes it is difficult to follow. It will be very helpful for the readers if a list of key abbreviations is provided.
It will be also helpful for the readers if the overall length of the manuscript is reduced.
Author Response
Dear Editors,
Please, see the attachment for the response to reviewer 2 comments.
Thanks

Reviewer 3 Report
The authors presented a well written and very comprehensive review of the history and current issues of the live attenuated rotavirus vaccines. They provided tables that enable readers to compare efficacy of different vaccine and factors that affect vaccine efficacy. They summarized factors that may determine the low protective efficacy of rotavirus vaccines in low-income countries so that researches who are working to improve rotavirus vaccines would have great overview of the issues that affect vaccine efficacy. It will be a very nice reference for anyone who is interested in rotavirus, vaccine developments.
Some minor issues.
Line 44: The most widely studied strain among these is rotavirus Simian agent 11 (RV SA11), which currently has 36 G-types and 51 P-types.
This sentence is confusion. It appears that SA11, not rotavirus in general, has 36 G-types and 51 P-types. Please modify
Line 296: Simian RRV (SA11) obtained from the serotype 3 isolated from 105 days old asymptomatic young Vervet rhesus monkey…
Please remove (SA11).
Line 399: Furthermore, infection with RRV-1 has been shown to induce monoclonal antibody against the VP4 of serotypes 3, 5, and 6 [142].
The study cited here used competition assay to demonstrate that antibodies induced in children immunized with RRV-1 can compete with previously described homotypic or heterotypic VP7 and VP4 neutralizing mAbs. This study did not directly isolate monoclonal antibodies from these children. Please modify the sentence.
Line 1488-1489: “a more significant efficacy of the RV1 against RVI especially after the first year of follow-up.”
Not clear what is “efficacy of the RV1 against RV1”. Please clarify.
Author Response
Dear Editors,
Please, see the attachment for the response to reviewer 3 comments.
Thanks

Round 2
Reviewer 1 Report
A much-improved version of the review. Substantial editing has been done with information put into tables.
The authors went to great length to justify the importance of the inclusion of so much.
Personally I would still like additional refinement is still needed as the tables are now immense. Simply moving the content to another form.
I do not feel this is suitable as a review article because of its length.
Author Response
The response to the reviewer's comment (Round 2) has been attached.
Please note that in the course of editing this article, additional statements/words are highlighted in yellow colour. Also, Tables 1, 5, 8, 9, and 10 are now in the supplementary list as Supplementary 1, 2, 3, 4, and 5.
POINT 1: A much-improved version of the review. Substantial editing has been done with information put into tables.
RESPONSE 1: The authors appreciated your encouraging comments. Thanks.
POINT 2: The authors went to great length to justify the importance of the inclusion of so much.
RESPONSE 2: Yes, thanks a lot for your understanding. As stated before, it is indeed an extensive review, which presented rotavirus vaccines with the broader view of their historical development, efficacies and challenges. Considerable efforts were also put in place to compare several reports of the vaccine efficiency from different parts of the world.
POINT 3: Personally, I would still like additional refinement is still needed as the tables are now immense. Simply moving the content to another form.
RESPONSE 3: Yes, this is predicted as well. The authors decided to move some of the tables to the supplementary documents while further editing was done. With this, 5 out of the 10 tables have been moved to the supplementary document. Few editing has been done as well, for example, in Table 1 and Section 6.12.
POINT 4: I do not feel this is suitable as a review article because of its length.
RESPONSE 4: The authors considered the publication of this time-relevant article in a review format for immediate widespread accessibility. Majority of the published books are not openaccess, unlike the MDPI journals. Furthermore, a book contains many sections, which must be collated and edited together before the final publication, and this may probably take a significant number of months. The authors pray that the reviewer, together with the editors, considers a journal-based publication for this review article for the following opinions:
√ the very bulk of the review article is now reduced to about 44 pages or less
√ half of the total number of tables has been moved to the supplementary document
√ half of the entire review is a cited reference, which reveals the in-depth and extensive review work put together by the authors
√ books are most likely not an open-access unlike the MDPI journals
√ open-access journal-based publication enhances rapid spread and dissemination of information
